# A developmental framework linking neurogenesis and circuit formation in the *Drosophila* CNS

**Brandon Mark[1], Sen-Lin Lai[1], Aref Arzan Zarin[1†], Laurina Manning[1], Heather Q Pollington[1], Ashok Litwin-Kumar[2], Albert Cardona[3], James W Truman[4], Chris Q Doe[1]***

[1]Institute of Neuroscience, Howard Hughes Medical Institute, University of Oregon, Eugene, United States; [2]Mortimer B Zuckerman Mind Brain Behavior Institute, Department of Neuroscience, Columbia University, New York, United States; [3]Janelia Research Campus, Howard Hughes Medical Institute, MRC Laboratory of Molecular Biology, Department of Physiology, Development & Neuroscience, University of Cambridge, Cambridge, United Kingdom; [4]Janelia Research Campus, Howard Hughes Medical Institute, Friday Harbor Laboratories, University of Washington, Friday Harbor, United States

**Abstract** The mechanisms specifying neuronal diversity are well characterized, yet it remains unclear how or if these mechanisms regulate neural circuit assembly. To address this, we mapped the developmental origin of 160 interneurons from seven bilateral neural progenitors (neuroblasts) and identify them in a synapse-scale TEM reconstruction of the *Drosophila* larval central nervous system. We find that lineages concurrently build the sensory and motor neuropils by generating sensory and motor hemilineages in a Notch-dependent manner. Neurons in a hemilineage share common synaptic targeting within the neuropil, which is further refined based on neuronal temporal identity. Connectome analysis shows that hemilineage-temporal cohorts share common connectivity. Finally, we show that proximity alone cannot explain the observed connectivity structure, suggesting hemilineage/temporal identity confers an added layer of specificity. Thus, we demonstrate that the mechanisms specifying neuronal diversity also govern circuit formation and function, and that these principles are broadly applicable throughout the nervous system.

**\*For correspondence:**
cdoe@uoregon.edu

**Present address:** [†]Department of Biology & Texas A & M Institute for Neuroscience (TAMIN), Texas A & M University, College Station, United States

## Introduction

Tremendous progress has been made in understanding the molecular mechanisms generating neuronal diversity in both vertebrate and invertebrate model systems. In mammals, spatial cues generate distinct pools of progenitors, which generate neuronal diversity in each spatial domain (*Sagner and Briscoe, 2019*). The same process occurs in invertebrates like *Drosophila*, but with a smaller number of cells, and this process is particularly well understood. The first step occurs when spatial patterning genes act combinatorially to establish single, unique progenitor (neuroblast) identities (*Skeath and Thor, 2003*). These patterning genes endow each neuroblast with a unique spatial identity (*Figure 1A*, left).

The second step is temporal patterning – the specification of neuronal identity based on birthorder – an evolutionarily conserved mechanism for generating neuronal diversity (*Kohwi and Doe, 2013*; *Rossi et al., 2017*). Here, we focus on *Drosophila* embryonic neuroblasts, which undergo a cascade of temporal transcription factors: Hunchback (Hb), Krüppel (Kr), Pdm, and Castor (Cas) (*Isshiki et al., 2001*). Each temporal transcription factor is inherited by ganglion mother cells (GMCs)

**Figure 1.** Mapping neurons with shared developmental origin in a transmission electron microscopy (TEM) reconstruction: clonally related neurons project widely and localize synapses to both sensory and motor neuropil. (A) Three mechanisms specifying neuronal diversity. Neuroblasts characterized here are shown in dark gray and arise from all anteroposterior and mediolateral positions of the neuroectoderm (dorsal view: anterior up, ventral midline at left of panel). They undergo temporal patterning as shown in the middle panel (posterior view: dorsal up). Nb: Numb; N: Notch; A1L:

*Figure 1 continued on next page*

*Figure 1 continued*

abdominal segment A1 left side. (**A′**) Schematic of newly hatched larval central nervous system (CNS) in a dorsal view, anterior up (top), or in a posterior view, dorsal up (bottom). All images in the figures are shown in posterior view, dorsal up unless noted otherwise. (**B**) Single neuroblast clones for the indicated neuroblasts (larval neuroblast names given in *Supplementary file 1*) each generated in A1L with *dpn(FRT.stop)LexA.p65* and assayed in newly hatched larvae. We recovered n > 2 clones for each newly characterized lineage; NB4-1 was previously characterized (*Lacin and Truman, 2016*). Dashed lines: neuropil border; vertical dash: midline. Scale bar, 10 μm. (**C**) The corresponding neurons traced in the TEM reconstruction. Dashed lines: neuropil border; vertical dash: midline. Arrows denote fascicles entering the neuropil; also shown in (**D**). (**D**) Each clone characteristically has either one or two fascicles entering the neuropil (black arrows, yellow highlight). (**E**) There are a similar number of neurons per neuroblast clone in A1L (left) and A1R (right). (**F**) Presynaptic and postsynaptic density maps (75% threshold) for each neuroblast lineage. Dashed lines: neuropil border; vertical dash: midline. All density maps are from neurons in A1L (cell bodies not shown). (**G**) Seven bilateral neuroblast lineages in segment A1 left traced in the TEM reconstruction. Inset: same projections, lateral view, anterior up. L: left; R: right; D: dorsal; V: ventral; A: anterior. (**H**) Summary.

born during each expression window. The combination of spatial and temporal factors endows each GMC with a unique identity (*Figure 1A*, middle).

The third step is hemilineage specification, which was initially characterized in *Drosophila* larval and adult neurogenesis (*Lee et al., 2020*; *Truman et al., 2010*), and may also be used in vertebrate neurogenesis (*Peng et al., 2007*). Hemilineages are formed by GMC asymmetric division into a pair of post-mitotic neurons; during this division, the Notch inhibitor Numb (Nb) is partitioned into one neuron (Notch$^{OFF}$ neuron), whereas the other sibling neuron receives active Notch signaling (Notch$^{ON}$ neuron), thereby establishing Notch$^{ON}$ and Notch$^{OFF}$ hemilineages (*Figure 1A*, right). In summary, three mechanisms generate neuronal diversity within the embryonic central nervous system (CNS): neuroblast spatial identity, GMC temporal identity, and neuronal hemilineage identity.

A great deal of progress has also been made in understanding neural circuit formation in both vertebrates and invertebrate model systems, revealing a multi-step mechanism. Neurons initially target their axons to broad regions (e.g., thalamus/cortex), followed by targeting to a neuropil domain (glomeruli/layer), and finally forming highly specific synapses within the targeted domain (*Kolodkin and Tessier-Lavigne, 2011*; *Moyle et al., 2021*).

Despite the progress in understanding the generation of neuronal diversity and the mechanisms governing axon guidance and neuropil targeting, how these two developmental processes are coordinated remains largely unknown. While it is accepted that the identity of a neuron is linked to its connectivity, the developmental mechanisms involved are unclear. For example, do clonally related neurons target similar regions of the neuropil due to the expression of similar guidance cues? Do temporal cohorts born at similar times show preferential connectivity? Here, we address the question of whether any of the three developmental mechanisms (spatial, temporal, hemilineage identity) are correlated with any of the three circuit-wiring mechanisms (neurite targeting, synapse localization, connectivity). We map the developmental origin for 80 bilateral pairs of interneurons in abdominal segment 1 (A1) by identifying and reconstructing these neurons within a full CNS TEM volume (*Ohyama et al., 2015*); this is over a quarter of the ~300 neurons per hemisegment. We make the unexpected observation that hemilineage identity determines neuronal projection to sensory or motor neuropils; thus, neuroblast lineages coordinately produce sensory and motor circuitry. In addition, we show that neurons with shared hemilineage-temporal identity target pre- and postsynapse localization to similar positions in the neuropil, and that hemilineage-temporal cohorts share more common synaptic partners than that produced by neuropil proximity alone. Thus, temporal and hemilineage identity plays essential roles in establishing neuronal connectivity.

## Results

### Mapping neuronal developmental origin in a TEM reconstruction

To relate developmental mechanisms to circuit establishment mechanisms, we first needed to identify the developmental origin of neurons within a TEM reconstruction of the larval CNS (*Ohyama et al., 2015*), allowing us to quantify neuronal projections, synapse localization, and connectivity. We took three approaches. First, we generated GFP-marked clones for seven different neuroblasts, representing different spatial positions in the neuroblast array (*Figure 1A*, left). We imaged each clone by light microscopy in newly hatched larvae – the same stage used for the TEM reconstruction – so that we could match clonal morphology at the light and TEM levels

(*Figure 1B, C*). All assayed neuroblast clones had a reproducible clonal morphology including the number of fascicles entering the neuropil, cell body position, and axon/dendrite morphology (*Figure 1B*; data not shown). We identified the parental neuroblast generating each clone by comparing clonal morphology to embryonic DiI single neuroblast clones (*Bossing et al., 1996*; *Schmid et al., 1999*; *Schmidt et al., 1997*), larval neuroblast clones (*Birkholz et al., 2015*; *Lacin and Truman, 2016*), and cell body position.

To identify each of the seven genetically labeled neuroblast clones in the TEM volume, we matched lineage-specific features present in both light and TEM analyses. We identified neurons that had clustered cell bodies, clone morphology matching that seen by light microscopy (*Figure 1C*), and one or two fascicles entering the neuropil (*Figure 1C, D*). The similarity in overall clone morphology between genetically marked clones and TEM reconstructed clones was obvious (compare *Figure 1B, C*). We validated these assignments using two methods. First, we used neuroblast-specific Gal4 lines (*Lacin and Truman, 2016*; *Seroka and Doe, 2019*) and multicolor flip out (MCFO) (*Nern et al., 2015*) to label individual neurons within each lineage. We found that in each case we could match the morphology of an MCFO-labeled single neuron from a known neuroblast to an identical single neuron in the same neuroblast clone within the TEM reconstruction (data not shown). Second, we reconstructed the same seven lineages in a second hemisegment, A1R. We observed similar neuron numbers, similar fascicles per clone, and similar clonal morphology (*Figure 1E*; data not shown). Thus, neuroblast lineages are highly stereotyped in left and right hemisegments.

We found that, while lineages have stereotyped and distinct morphology, they all had broad projections within the neuropil. We mapped the distribution of pre- and postsynapses for each neuroblast clone and found that, consistent with neuronal morphology, synapses were distributed widely across the neuropil (*Figure 1F*). We conclude that clonally related neurons project widely and localize synapses widely (*Figure 1G, H*).

In total, we have mapped 14 neuroblast clones in the TEM volume (seven in A1L, seven in A1R). These lineages contain 160 interneurons (80 each in A1l and A1r), containing 7794 presynapses and 19,468 postsynapses. We also include the previously traced sensory afferents and dendrites from all 32 motor neurons (*Jovanic et al., 2016*; *Ohyama et al., 2015*; *Zarin et al., 2019*). All data are publicly available from https://github.com/bjm5164/Mark2020_larval_development (copy archived at swh:1:rev:43e0a22c5381427aa6670c55ec4de76f5ad39568; *Mark, 2020*). We note that some of the earliest born neurons are not included either because their cell bodies are in contact with the neuropil and they do not fasciculate with clonal fascicles, precluding assignment to a specific neuroblast lineage, or they do not maintain marker expression at larval hatching. This can sometimes, but not always, lead to a gap between the deepest mapped neuron and the neuropil (*Figure 1C*). The morphology and function of the earliest born neurons will be described elsewhere. Nevertheless, the current data is sufficiently comprehensive for mapping developmental mechanisms to circuit assembly mechanisms.

In the following sections, we first analyze the relationship of developmental mechanisms to neuronal projections and synapse localization within the neuropil; we conclude by exploring the relationship between developmental mechanisms and neuronal connectivity.

## Individual lineages generate two distinct morphological classes of neurons

Perhaps the most important neuropil subdivision in both mammalian spinal cord and *Drosophila* ventral nerve cord is the segregation of motor and sensory processing domains. In *Drosophila*, motor dendrites target a dorsal region of the neuropil, while sensory neurons target a ventral region of the neuropil (*Landgraf et al., 2003*; *Mauss et al., 2009*; *Zarin et al., 2019*; *Zlatic et al., 2009*). We extend these findings to show that premotor neurons also target pre- and postsynapses to the dorsal neuropil, and targets of sensory afferents target pre- and postsynapses to the ventral neuropil (*Figure 2A–C*). Thus, the neuropil has an important functional subdivision along the dorsoventral axis.

Upon examination of each lineage, we found that nearly all neuroblasts made two morphologically distinct types of neurons (except NB2-1; *Figure 2—figure supplement 1*). We used NBLAST (*Costa et al., 2016*) to quantify the morphological similarity of neurons within each neuroblast lineage and found that most lineages made two distinct morphological classes of neurons (or one class

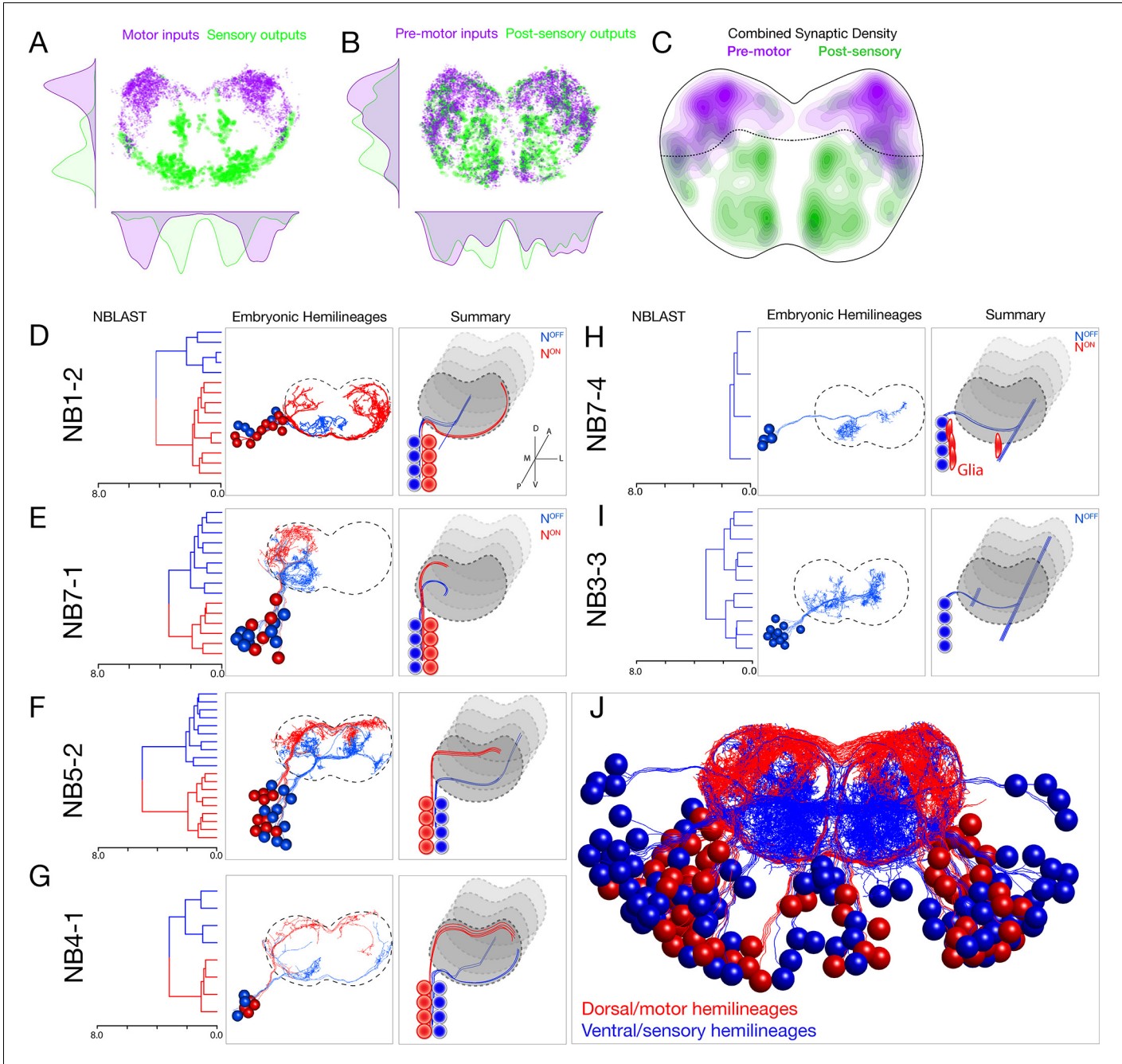

**Figure 2.** Each neuroblast lineage generates neurons projecting to dorsal/motor or ventral/sensory neuropil. (A–C) Organization of motor and sensory domains within the ventral nerve cord neuropil. (A) Motor neuron postsynapses (purple) and sensory neuron presynapses (green) showing dorsoventral segregation. Plots are 1D kernel density estimates for dorsoventral or mediolateral axes. Purple dots represent a single postsynaptic site. Green dots represent a single presynaptic site scaled by the number of outputs from that presynaptic site. (B) Premotor neuron postsynaptic sites (>3 synapses onto a motor neuron in segment A1) or postsensory neuron presynaptic sites (pre >3 synapses with an A1 sensory neuron) show that connecting neurons are still restricted to dorsal or ventral neuropil domains. (C) 2D kernel density estimates of all presynaptic and postsynaptic sites for premotor and postsensory neurons outline the regions of sensory (green) and motor (purple) processing in the ventral nerve cord. (D–I) NBLAST clustering for the indicated neuroblast progeny typically reveals two morphological groups of neurons or glia (red/blue) that project to dorsal or ventral neuropil; these are candidate hemilineages. Cluster cutoffs were set at 3.0 for all lineages. (J) Superimposition of all dorsal candidate hemilineages (red) and all ventral candidate hemilineages (blue). Data from NBs 1-2, 3-3, 4-1, 5-2, 7-1, and 7-4 (this figure) and NB2-1 (*Figure 2—figure supplement 1*).

The online version of this article includes the following figure supplement(s) for figure 2:

**Figure supplement 1.** NB2-1 has Notch[ON] and Notch[OFF] neurons, but all project to a similar neuropil domain.

**Figure supplement 2.** Clonally related neurons can have different morphology.

of neurons plus glia). Strikingly, each neuronal class projected to the dorsal/motor neuropil or the ventral/sensory neuropil (*Figure 2D–H*). Note that NB3-3 had only one fascicle at the NBLAST threshold used, and all neurons were ventral-projecting (*Figure 2I*). These classes were more distinct from one another than from neurons in other lineages (*Figure 2—figure supplement 2*). We conclude that most lineages generate two distinct classes of neurons that target either the ventral/sensory or dorsal/motor neuropils (*Figure 2J*).

## Hemilineages produce sensory and motor processing neurons via a Notch-dependent mechanism

Recent work has shown that within the larval and adult CNS most neuroblast lineages generate Notch$^{ON}$ neurons with a similar clonal morphology (called the Notch$^{ON}$ hemilineage), and Notch$^{OFF}$ hemilineage with a different morphology (*Harris et al., 2015*; *Lacin and Truman, 2016*; *Lee et al., 2020*; *Truman et al., 2010*). We hypothesized that the observed morphological differences within our seven embryonic neuroblast lineages may be due to hemilineage identity. This hypothesis was strengthened by the fact that the sole neuroblast lineage that generated a single morphological class, NB3-3, had previously been shown to make only one 'Notch$^{OFF}$' hemilineage via type 0 divisions (*Baumgardt et al., 2014*; *Wreden et al., 2017*). Based on these data, we hypothesize that each neuroblast makes a Notch$^{ON}$ hemilineage projecting to the dorsal motor neuropil (or glia), and a Notch$^{OFF}$ hemilineage projecting to the ventral sensory neuropil.

To test this hypothesis, we took two approaches: first, we generated a lineage-specific reporter for Notch$^{ON}$ neurons to see if Notch$^{ON}$ neurons in specific neuroblast linages always project to the dorsal neuropil; and second, we tested whether misexpression of constitutively active Notch can redirect ventral projections into the dorsal neuropil. To generate a neuroblast lineage-specific Notch$^{ON}$ reporter, we CRISPR engineered the Notch target gene *hey*, placing a *T2A:FLP* exon in frame with the terminal *hey* exon at the endogenous *hey* locus. Combining this fly with a neuroblast-specific Gal4 line (*NB5-2-gal4* or *NB7-1-gal4*) plus a FLP-dependent reporter (*UAS-FRT-stop-FRT-myr:sfGdP:HA*) resulted in myr:GdP:HA expression specifically in Notch$^{ON}$ neurons within a neuroblast lineage. This resulted in specific labeling of dorsal projecting neurons within the NB5-2 lineage (*Figure 3A, A'*) or the NB7-1 lineage (*Figure 3B, B'*). We conclude that Notch$^{ON}$ neurons specifically project to the dorsal motor neuropil.

We next asked whether Notch activity determines dorsal/ventral neuropil projections. We used Gal4 lines specifically expressed in single neuroblast lineages (NB1-2, NB5-2, NB7-1, or NB7-4) (*Lacin and Truman, 2016*; *Seroka and Doe, 2019*) to misexpress a constitutively active form of Notch (Notch$^{intra}$) in individual neuroblast lineages. Wild-type lineages had both dorsal and ventral projections (*Figure 3E*) or glial and ventral projections (*Figure 3F*). In contrast, Notch expression throughout the NB1-2, NB5-2, or NB7-1 lineages led to a re-routing of projections from the ventral neuropil to the dorsal neuropil (*Figure 3G–I*). Notch expression throughout the NB7-4 lineage led to a loss of ventral projecting neurons and an increase in glia (*Figure 3J*). In addition, we note that ascending and descending projection neurons are normally generated by ventral Notch$^{OFF}$ hemilineages (*Figure 3—figure supplement 1*). Importantly, these ascending and descending projection neurons were completely lost following Notch$^{intra}$ expression (*Figure 3G–I*, insets). These results are summarized in *Figure 3K*. Our observation that Notch$^{OFF}$ hemilineages make more complex and lengthy neurons than the Notch$^{ON}$ hemilineages is similar to that observed for larval brain lineages (*Lee et al., 2020*), suggesting that this is a conserved mechanism. The Notch$^{intra}$ phenotypes we observed are likely due to a Notch$^{OFF}$ to Notch$^{ON}$ hemilineage transformation, rather than death of ventral projecting neurons, as we observed the same number of neurons in control versus Notch$^{intra}$ embryos at stage 16/17 (NB1-2 control: 12.2, n = 10 hemisegments; NB1-2 Notch$^{intra}$: 14.1, n = 12 [p=0.0785]; NB5-2 control: 20.3, n = 16 hemisegments; NB5-2 Notch$^{intra}$: 19.3, n = 26 [p=0.2311]; NB7-1 control: 18.9, n = 7 hemisegments; NB7-1 Notch$^{intra}$: 19.8, n = 11 [p=0.6166]; NB7-4 control: 8.6, n = 9 hemisegments; NB7-4 Notch$^{intra}$: 9.5, n = 11 [p=0.1710]). In conclusion, we show that NBLAST can be used to accurately identify neuroblast hemilineages; that Notch$^{ON}$/Notch$^{OFF}$ hemilineages project to motor/sensory neuropil domains, respectively; and most importantly, that hemilineage identity determines neuronal targeting to the motor or sensory neuropil. Thus, each neuroblast coordinately generates similar numbers of sensory and motor processing neurons throughout its embryonic lineage.

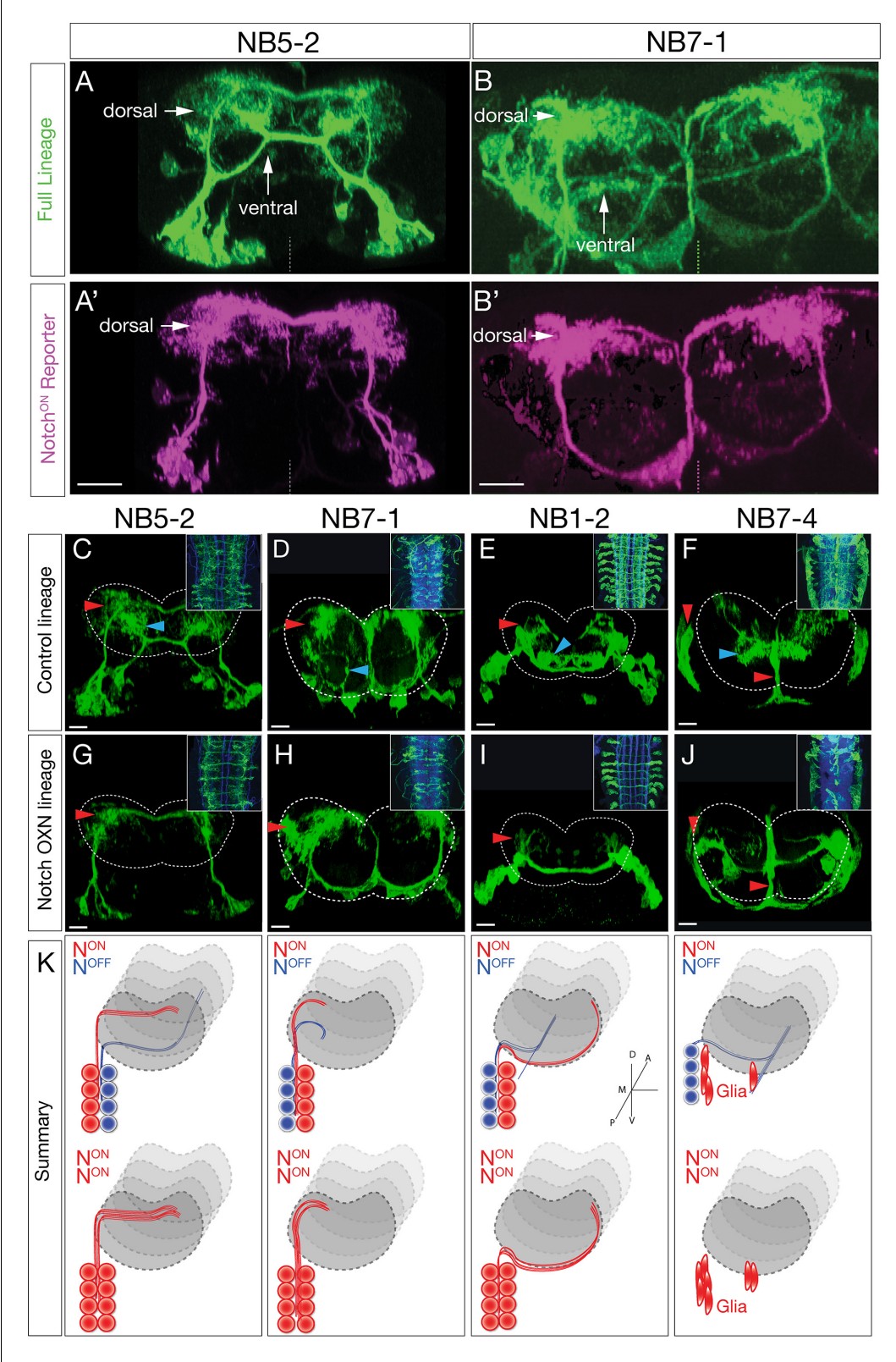

**Figure 3.** Hemilineage identity drives neuronal projections to motor or sensory neuropil. (**A, B**) A Notch reporter (*hey:T2A:FLP, UAS-myr:GFP, UAS-FRT-stop-FRT-myr:sfGdP:HA*) specifically labels dorsal projections within the indicated neuroblast lineages. Myr:GFP labels the whole lineage (green), myr:sfGdP:HA is a myristoylated (myr) superfolder green dead protein (sfGdP) fused to hemoagglutinin (HA), which labels the Notch[ON] hemilineage (magenta). (**A', B'**) Staining for HA reveals the Notch[ON] neuron projections. Vertical dashes: midline; dorsal: up. (**C–F**) Control neuroblast lineages

*Figure 3 continued on next page*

*Figure 3 continued*

project to both dorsal neuropil (red arrowheads) and ventral neuropil (cyan arrowheads). Scale bars, 10 µm. (G–J) Lineage-specific Notch[intra] expression transforms ventral projections to dorsal projections or glia (red arrowheads). Cell numbers in control and Notch misexpression are similar (see text). n > 3 for all experiments. Dashed lines: neuropil border; dorsal: up. Scale bars, 5 µm. (K) Summary.

The online version of this article includes the following figure supplement(s) for figure 3:

**Figure supplement 1.** Ventral hemilineages have projection neurons.

## Hemilineages target synapses to subdomains of motor or sensory neuropil

To identify a relationship between hemilineage identity and synapse localization, we mapped the pre- and postsynapse localization for 12 bilateral hemilineages (24 total) in segment A1. Note that we show synapses from both A1L and A1R neuroblast lineages, which highlights the similarity and stereotypy of synapse localization between left and right lineages (*Figure 4A*). We found that the neurons in dorsal hemilineages localized both pre- and postsynaptic sites to the dorsal/motor neuropil, whereas neurons in ventral hemilineages localized both pre- and postsynaptic sites to the ventral/sensory neuropil (*Figure 4A*). Consistent with these observations, we found that the vast majority of sensory output was to ventral hemilineages, and the vast majority of input to motor neurons was from dorsal hemilineages (*Figure 4B*). We conclude that, at least for the assayed hemilineages, Notch[ON] hemilineages target projections and pre- and postsynapses to the motor neuropil, whereas Notch[OFF] hemilineages target projections and pre- and postsynapses to the sensory neuropil (*Figure 4H*).

After showing that hemilineages target synapses to dorsal or ventral neuropil, we asked if individual hemilineages target distinct regions of the neuropil or if they have overlapping territories. We mapped the pre- and postsynaptic position for both dorsal and ventral hemilineages (excluding the NB7-4 glial hemilineage). We found that presynapses were localized to distinct regions of the neuropil (*Figure 4C, D*). Similarly, postsynapses were localized to distinct but slightly more overlapping regions of the neuropil (*Figure 4E, F*). We quantified synapse similarity (a measure of the average distance between synapses of two neurons) using published methods (*Schlegel et al., 2016*). We found that synapses from lineages in A1L and A1R had similar relative positions in the neuropil (*Figure 4—figure supplement 1A–D*), highlighting the stereotypy of synapse localization. Importantly, neurons in a hemilineage showed greater similarity in synaptic positions (pre or post) than unrelated neurons (*Figure 4G*, *Figure 4—figure supplement 1E, F*). We conclude that each hemilineage targets its presynapses (and to a lesser extent postsynapses) to small domains of the sensory or motor neuropil (*Figure 4H*), strongly suggesting that the developmental information needed for neuropil targeting is shared by neurons in a hemilineage, but not by all neurons in a complete neuroblast lineage (see Discussion).

## Mapping temporal identity in a TEM reconstruction: radial position is a proxy for neuronal birth-order

To investigate the role of temporal identity in determining neuronal projections, synapse localization, or connectivity, we needed to identify the temporal identity of all 160 interneurons analyzed here. We used two methods. First, we confirmed that temporal transcription factors (Hb, Kr, Pdm, and Cas) have a radial distribution in the embryonic CNS, with early-born Hb+ neurons positioned in a deep layer adjacent to the developing neuropil and late-born Cas+ neurons are at the most superficial position within the CNS (*Isshiki et al., 2001*; *Kambadur et al., 1998*; *Figure 5A*). Importantly, we show that this distribution persists for more stable Hb and Cas reporters in the newly hatched larval CNS (*Figure 5B, C*), the stage of the TEM reconstruction. Thus, radial position can be used as a proxy for temporal identity in both embryos and newly hatched larvae (*Figure 5D*). Second, we used MCFO to identify additional single Hb+ or Cas+ neurons and matched them to the morphologically identical neuron in the TEM volume (*Figure 5E ,F*, *Figure 5—figure supplement 1*; data not shown). In total, we identified 54 Hb+ neurons and 44 Cas+ neurons within the TEM volume. We measured their cortex neurite length (neurite length from the cell body to the neuropil entry point; i.e., the length of the neurite as it traverses the cellular cortex) and found that experimentally verified Hb + neurons were closer to the neuropil, whereas Cas+ neurons were further from the neuropil

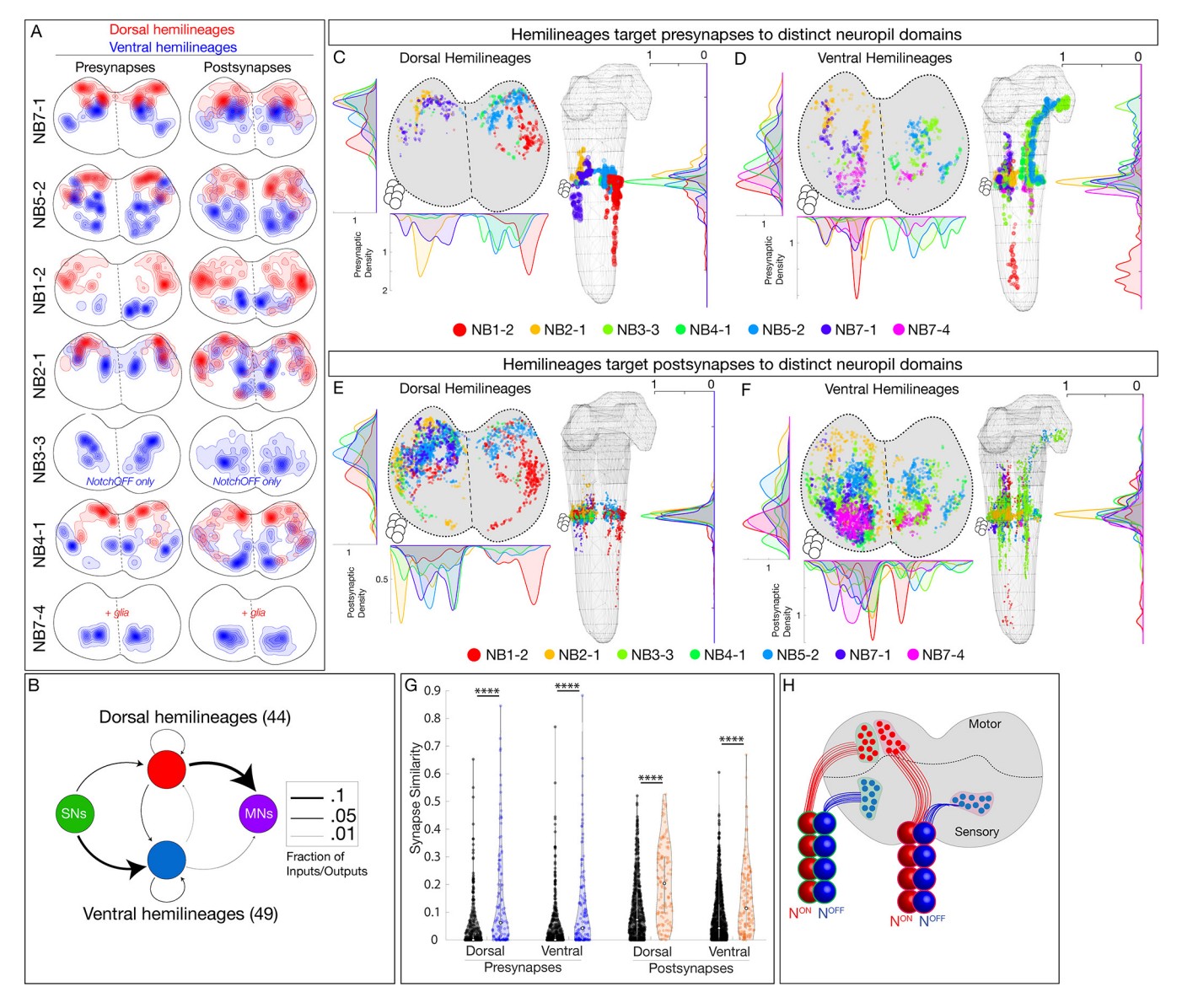

**Figure 4.** Hemilineages target pre- and postsynapses to subdomains of dorsal or ventral neuropil. (**A**) Density maps for dorsal synapses (red) or ventral synapses (blue) within each lineage. Dorsal hemilineages localize both presynapses and postsynapses to dorsal neuropil, whereas ventral hemilineages localize both presynapses and postsynapses to ventral neuropil. Vertical dashes: midline. Synapses from both A1L and A1R are shown to highlight left/right stereotypy. Synapses outside of segment A1 were from this analysis given the change in shape and orientation of the neuropil in more posterior segments and the central brain. See **Figure 4—figure supplement 1** for the bounds used. (**B**) Connectivity diagram showing sensory neurons provide inputs to neurons in ventral hemilineages, while motor neurons preferentially receive inputs from neurons in dorsal hemilineages. Edges represent fractions of outputs for sensory neurons, and fraction of inputs for motor neurons. (**C, D**) Presynaptic distributions of the indicated dorsal or ventral hemilineages from A1L. Dots represent single presynaptic sites with their size scaled by the number of associated (polyadic) postsynaptic sites. Circles: location of cell bodies. Note that NB1-2 ventral hemilineage presynapses (red dots) are located ventrally, but are not shown in the A1 cross-sectional view due to their position in posterior segments of the ventral nerve cord (VNC). (**E, F**) Postsynaptic distributions of the indicated dorsal or ventral hemilineages from A1L. Dots represent single postsynaptic sites. Circles: location of cell bodies. (**G**) Mean inter- versus intra-hemilineage synapse similarity scores for dorsal and ventral hemilineages show intra-hemilineage presynapse (blue) and postsynapse (orange) similarity is greater than inter-hemilineage (black) similarity. In this case, intra-hemilineage similarity represents comparisons only to neurons targeting the same region of the neuropil (dorsal-dorsal/ventral ventral). Error bars represent SEM. p<0.05 in all cases (Wilcoxon rank-sum test). (**H**) Summary.

The online version of this article includes the following figure supplement(s) for figure 4:

**Figure supplement 1.** Neurons in a hemilineage cluster their synapses to specific regions of the neuropil.

**Figure supplement 2.** Lineage-related neurons segregate their synapses on the basis of hemilineage.

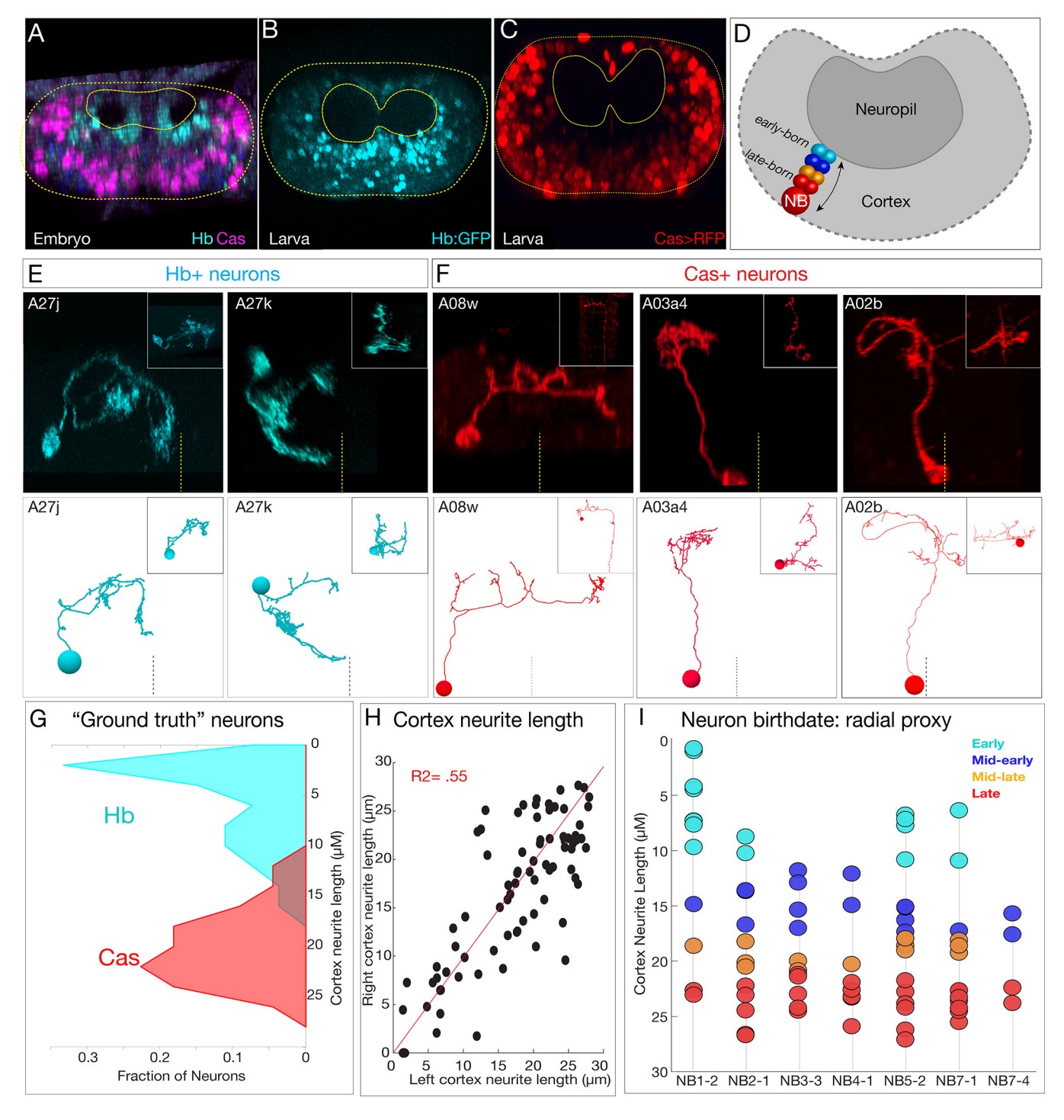

**Figure 5.** Mapping temporal identity in the TEM reconstruction: radial position is a proxy for neuronal birth-order. (**A–C**) Early-born Hb+ neurons are near the neuropil and late-born Cas+ neurons are far from the neuropil in the late embryo (**A**) and newly hatched larvae (**B, C**); larval reporters are recombineered *Hb:GFP* and *cas-gal4 UAS-RFP*. (**D**) Schematic showing correlation between radial position and early- or late-born temporal identity. (**E, F**) Examples of experimentally verified 'ground truth' early-born Hb+ neurons and late-born Cas+ neurons, identified using multicolor flip out with *cas-gal4* or using a CRISPR engineered *hb* locus where LexA:T2A is in frame with start of the *hb* coding sequence (see Materials and methods); additional examples in *Figure 5—figure supplement 1*. (**G**) Fraction of experimentally validated Hb+ or Cas+ 'ground truth' neurons at the indicated distance from the neuropil (cortex neurite length). n = 47 Cas+ neurons and 55 Hb+ neurons from segments T3-A2. (**H**) Cortex neurite lengths between the same neuron in the left and right hemisegment, showing that radial position is highly stereotyped. (**I**) Assignment of temporal identity based on

*Figure 5 continued on next page*

*Figure 5 continued*

radial position for neurons within the TEM reconstruction, calculated as the mean left/right distance for the same neuron in A1L and A1R. Note that not all lineages have all temporal cohorts, mirroring experimental observation that some neuroblast lineages do not express all temporal transcription factors (*Benito-Sipos et al., 2010*; *Cui and Doe, 1992*; *Isshiki et al., 2001*; *Tsuji et al., 2008*).

The online version of this article includes the following figure supplement(s) for figure 5:

**Figure supplement 1.** Ground truth Hb+ and Cas+ neurons used to define radial position as a proxy for temporal identity.

**Figure supplement 2.** NB1-2 dorsal hemilineage: radial proxy is unreliable for determining the temporal identity.

**Figure supplement 3.** Neurons with a common temporal identity project widely within the neuropil.

---

(*Figure 5G*). We conclude that neuronal radial position can be used as a proxy for neuronal temporal identity.

To determine the temporal identity of neurons within the seven bilateral neuroblast lineages, we measured cortex neurite length for each neuron. Importantly, the same neuron in A1L and A1R had similar cortex neurite lengths (*Figure 5H*), showing that cell body radial position was reproducible. We assigned 70 interneurons to one of four temporal cohorts (early, mid-early, mid-late, and late born) based on radial position (*Figure 5I*). We note that some of the earliest born neurons are not included (see Materials and methods), and we excluded the NB1-2 dorsal hemilineage from radial analysis as we found it to be an unreliable measure of birth-order for that hemilineage (see *Figure 5—figure supplement 2*). We conclude that cortex neurite length can be used as a proxy for temporal identity, that it is reproducible across at least two hemisegments, and that it can be used to approximate the temporal identity of any neuron in the TEM reconstruction. We use these temporal cohorts to explore the relationship between temporal identity and neuronal projections, synapse localization, and connectivity in the following sections.

After determining the temporal identity for 140 neurons, we asked whether neurons with a shared temporal identity had similar axon/dendrite projections or synapse localization. We found that neurons in each temporal class had broad neuronal projections, and no greater than unrelated neurons synapse similarity (*Figure 5—figure supplement 3*). We conclude that temporal identity, on its own, does not confer shared neuronal projections or synapse localization.

## Hemilineage-temporal cohorts have distinct synapse localization within the neuropil

Although temporal identity alone did not correlate with axon projections or synapse localization, it still could play a role in restricting synapse localization within individual hemilineages. Here, we analyze neurons in a hemilineage that share a temporal identity, subsequently called hemilineage-temporal cohorts (*Supplementary file 1*). We mapped presynaptic and postsynaptic localization for 20 hemilineage-temporal cohorts from seven neuroblasts, as illustrated for the NB5-2 dorsal hemilineage and the NB3-3 ventral hemilineage (*Figure 6A, B*) and the remainder of the hemilineage-temporal cohorts (*Figure 6—figure supplements 1* and *2*). We found that NB5-2 dorsal hemilineage-temporal cohorts target their *presynapses* to mostly non-overlapping neuropil domains (*Figure 6A*), whereas the NB3-3 hemilineage-temporal cohorts target their *postsynapses* to mostly non-overlapping neuropil domains (*Figure 6B*). When we expand this analysis to all hemilineages, we found that (a) hemilineage-temporal cohorts had more similar synaptic positions than hemilineage alone, and (b) dorsal hemilineage-temporal cohorts preferentially clustered presynapses, whereas ventral hemilineage-temporal cohorts preferentially clustered postsynapses (*Figure 6C–F*). This latter observation may reflect the need to precisely target premotor output to specific regions of motor neuron dendrites and the need to precisely receive sensory input from distinct sensory modalities (see Discussion). Our results also suggest that dorsal and ventral hemilineages utilize temporal identity in different ways to specify the targeting of either presynapses or postsynapses. We conclude that temporal identity subdivides hemilineages into smaller populations of neurons that target both projections and synapses to distinct subdomains within the larger hemilineage targeting domain (*Figure 6G*). Thus, hemilineage identity provides coarse targeting within the neuropil, and temporal identity refines targeting to smaller subdomains.

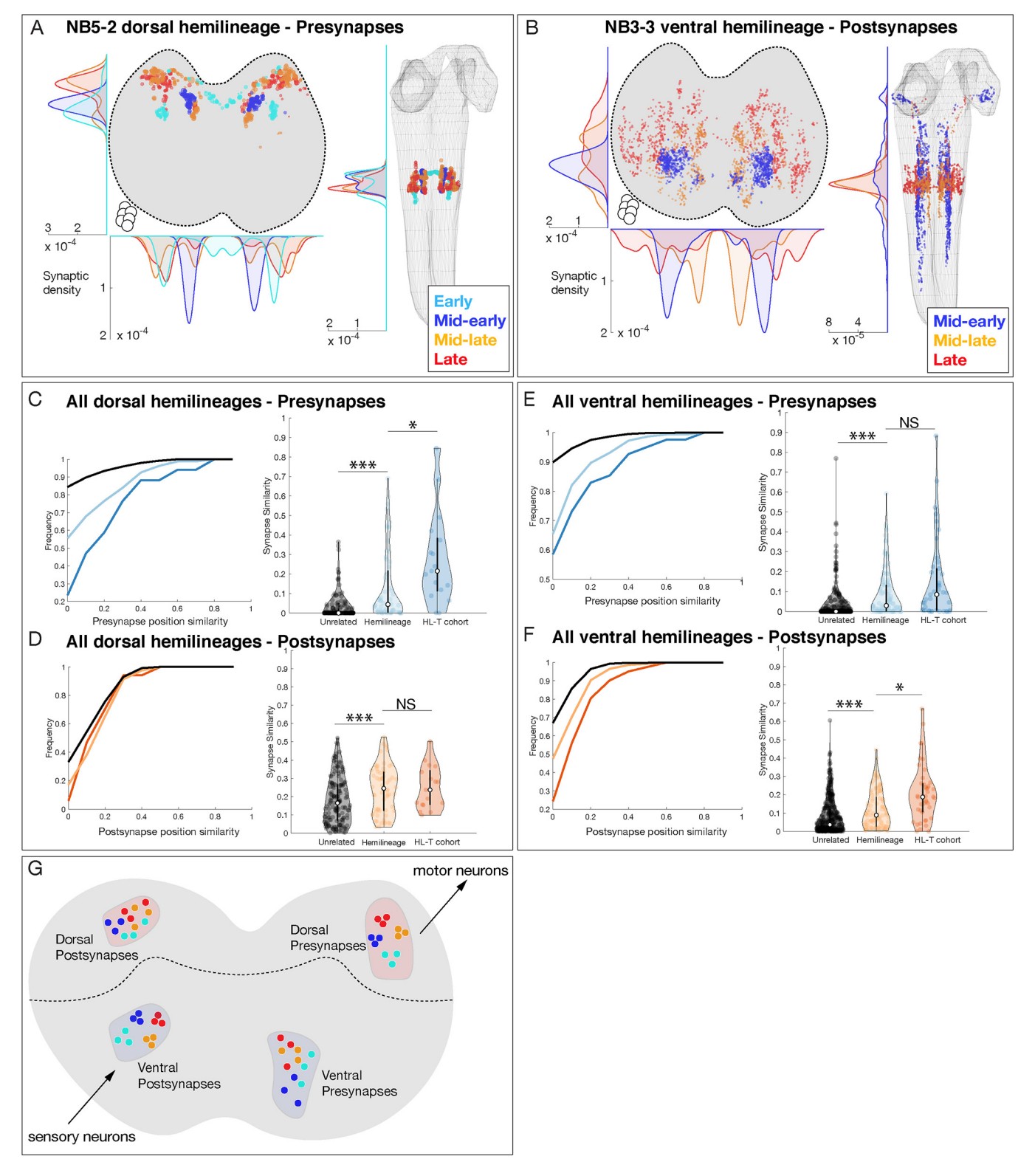

**Figure 6.** Hemilineage-temporal (HL-T) cohorts show synaptic tiling within motor or sensory neuropil. (**A, B**) Hemilineages target their synapses within the neuropil. (**A**) NB5-2 dorsal hemilineage neurons segregate their presynapses to distinct regions of the neuropil. (**B**) NB3-3 ventral hemilineage neurons segregate their postsynapses to distinct regions of the neuropil. Circles: location of cell bodies. (**C–F**) HL-T cohorts target their synapses within each hemilineage domain. (**C, D**) All dorsal hemilineages in A1L. Left: cumulative plots showing that hemilineages have greater synaptic clustering than

*Figure 6 continued on next page*

*Figure 6 continued*

unrelated neurons, but that HL-T cohorts have greater presynaptic clustering than hemilineages alone. Right: histograms showing that hemilineages have greater presynaptic clustering than unrelated neurons, but that HL-T cohorts have more presynaptic clustering than hemilineages alone. Circles: location of cell bodies. (E, F) All ventral hemilineages in A1L. Left: cumulative plots showing that hemilineages have greater postsynaptic clustering than unrelated neurons, but that HL-T cohorts have greater postsynaptic clustering than hemilineages alone. Right: histograms showing that hemilineages have greater postsynaptic clustering than unrelated neurons, but that HL-T cohorts have more postsynaptic clustering than hemilineages alone. Hemilineage similarity refers to hemilineage-related neurons from different temporal cohorts. *p<0.05 or ***p<0.001 (Wilcoxon rank-sum test). Error bars, SEM. (G) Summary. Dorsal hemilineages tend to cluster presynapses, which provide input to motor neurons; ventral hemilineages tend to cluster postsynapses, which receive input from sensory neurons.

The online version of this article includes the following figure supplement(s) for figure 6:

**Figure supplement 1.** Hemilineage-temporal cohort presynapse distribution in the neuropil.
**Figure supplement 2.** Hemilineage-temporal cohort postsynapse distribution in the neuropil.

## Hemilineage-temporal cohorts have shared connectivity

Work from the Heckscher lab has shown that early-born and late-born temporal cohorts in the NB3-3 lineage have unique connectivity and participate in escape or proprioception locomotor circuits, respectively (*Wreden et al., 2017*). This has led them to hypothesize that other hemilineage-temporal cohorts may have shared connectivity (*Meng and Heckscher, 2021*). Below, we test this hypothesis for 20 hemilineage-temporal cohorts from seven different neuroblasts (*Supplementary file 1*). Here, we compare connectivity of hemilineage-temporal cohorts to unrelated neurons, neurons sharing a temporal identity, and neurons sharing a hemilineage identity. This allows us to test the hypothesis that hemilineage-temporal cohorts have more shared connectivity compared to other developmental groupings. First, we analyzed the connectome of 12 hemilineages plus the motor and sensory neurons in segment A1. In total, we analyzed 160 interneurons, 56 motor neurons, and 86 sensory neurons, which corresponded to approximately 25% of all inputs and 14% of all outputs for the 12 hemilineages. We found similar connectivity for the same neurons on the left and right side of the segment (*Figure 7A–C*), validating the accuracy of the neuronal reconstructions as well as the stereotypy of neurogenesis. Consistent with anatomical observations, we found that neurons in a hemilineage were clustered together in network space, suggesting that hemilineage-related neurons are highly interconnected and functionally distinct (*Figure 7D–F*), similar to what has been suggested previously (*Harris et al., 2015*).

Next, to determine if temporal identity could provide additional specificity to the observed hemilineage connectivity, we compared the connectivity of hemilineages to hemilineage-temporal cohorts. Here, we focus specifically on the 160 interneurons we have traced in the TEM volume, excluding motor and sensory neurons. We quantified the average connectivity distance between neurons related by hemilineage or hemilineage-temporal cohort. Neuron pairs that are directly connected have a network distance of one synapse; neurons that share a common input or output have a network distance of two synapses, up to an observed maximum of seven synapses. Importantly, neurons within a hemilineage had a lower network distance (i.e., greater connectivity) than unrelated neurons, while neurons within a hemilineage-temporal cohort had a lower network distance than those related by hemilineage alone (*Figure 7G–J*). Interestingly, we found that the largest difference between hemilineage and hemilineage-temporal cohort connectivity was at the two synapse distance. While less than 20% of hemilineage-related or hemilineage-temporal cohort-related neurons were directly connected, 75% of hemilineage-temporal cohort neurons were separated by two synapses or less (*Figure 7I, J*). We conclude that hemilineage-temporal cohorts share common connectivity and provide increased partner specificity compared to connectivity of hemilineages alone (*Figure 7K*).

## Temporal cohorts have greater shared connectivity with each other than predicted by neuropil proximity alone

Next, we explored the relationship between temporal identity and connectivity. Peter's rule proposes that connectivity is determined primarily by the overlap of axons and dendrites (*Rees et al., 2017*), raising the possibility that developmental mechanisms confer connectivity specificity simply by targeting sets of neurons to appropriate regions of the neuropil. To determine the role of

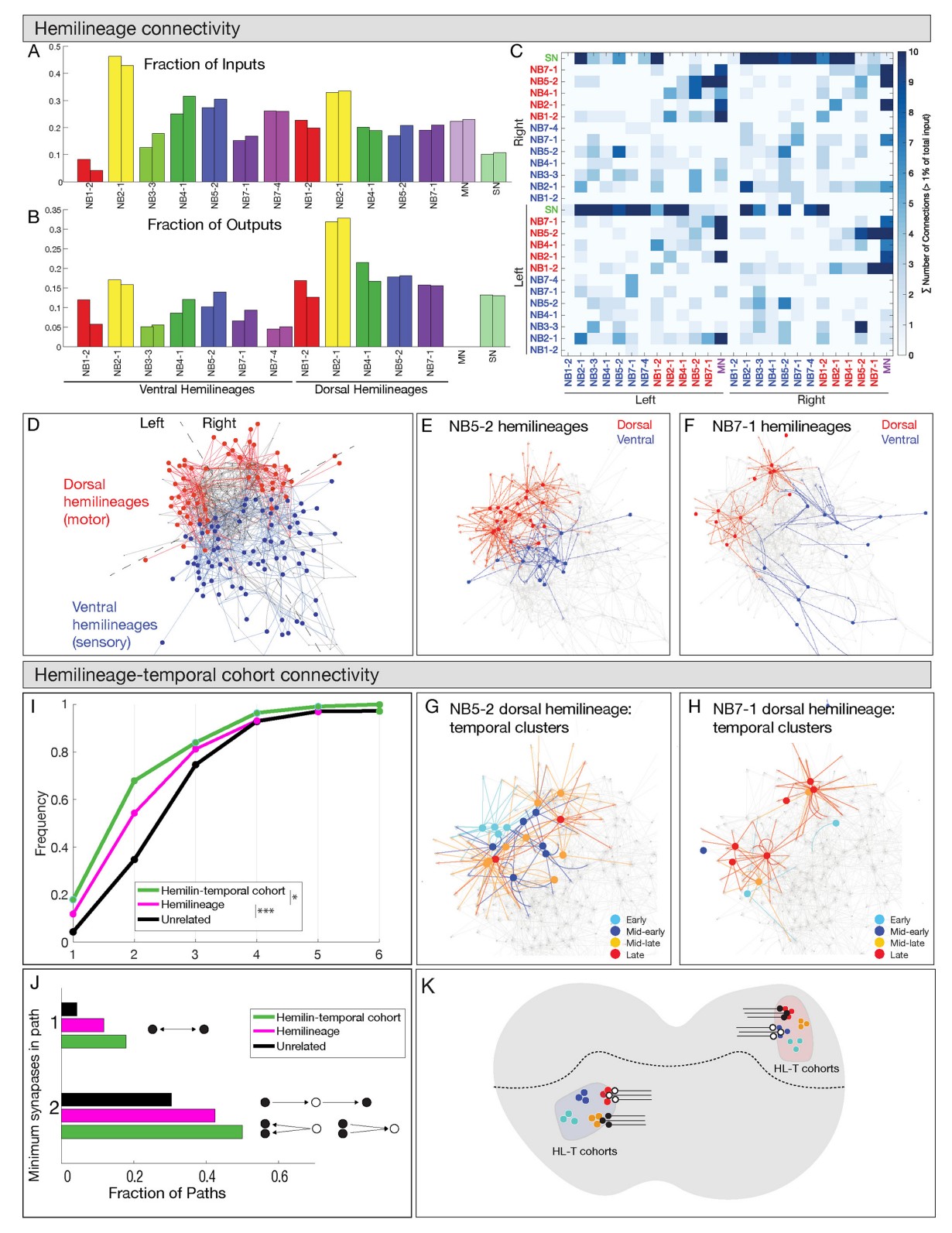

**Figure 7.** Hemilineages and hemilineage-temporal cohorts have more shared connectivity than unrelated neurons. (**A–C**) Connectivity is similar for hemilineages in A1L and A1R. (**A, B**) Fraction of inputs/outputs for each hemilineage; (**C**) heatmap showing connectivity between dorsal hemilineages (red), ventral hemilineages (blue), motor neuron dendrites (MN), and sensory neuron axons (SN). (**D–F**) Dorsal and ventral hemilineages have distinct connectivity. (**D**) Force directed network graph of all 160 interneurons, together with sensory afferents and motor dendrites. Neurons with similar

*Figure 7 continued on next page*

*Figure 7 continued*

connectivity appear closer in network space. Red edges represent dorsal hemilineage connectivity; blue edges represent ventral hemilineage connectivity. (E) Force-directed network graph highlighting the lack of shared connectivity between dorsal and ventral hemilineages in the NB5-2 progeny. (F) Force-directed network graph highlighting the lack of shared connectivity between dorsal and ventral hemilineages in the NB7-1 progeny. (G–J) Hemilineage-temporal cohorts within a hemilineage have shared connectivity. (G, H) Force-directed network graphs highlighting the shared connectivity of hemilineage-temporal cohorts within NB5-2 or NB7-1 progeny. (I) Cumulative distribution of the number of synapses between temporal cohorts of hemilineage-related neurons, hemilineage-related neurons, or random neurons. Neurons that belonged to a temporal cohort with only one neuron were not analyzed (16 neurons). Random neurons were selected from the same hemisegment. (J) Quantification of the number of directly connected pairs of neurons, neurons separated by one or two synapses. Black circles represent pairs of neurons connected by one synapse (top) or two synapses (bottom). (K) Summary.

neuropil proximity in connectivity, we calculated the spatial overlap of presynapses and postsynapses for all pairwise neuronal combinations, which ranged from no overlap to high overlap (*Figure 8A*, schematic). This measurement is analogous to previous synapse similarity measures, but instead we compared the presynapse positions of one neuron to the postsynapse positions of another. The majority of neurons have very little spatial overlap between presynapses and postsynapses (*Figure 8A'*). Interestingly, even neurons with the highest observed levels of overlap were not always connected (*Figure 8A''*). Thus, proximity alone cannot explain the observed connectivity, consistent with a role for hemilineage-temporal cohorts providing increased synaptic specificity.

To test the hypothesis that hemilineage-temporal identity confers synaptic specificity, we shuffled the connectivity by preserving the output degree of each neuron while setting the connection probability as a function of pre/post synaptic overlap. We found that hemilineage-temporal cohort to hemilineage-temporal cohort connectivity (*Figure 8B*, red dashed line) was greater than unrelated neuron wiring (*Figure 8B*, black) or proximity-based wiring (*Figure 8B*, cyan and magenta).

Finally, we asked if there was greater connectivity between hemilineage-temporal cohorts compared to unrelated neurons, neurons from the same temporal cohort, or neurons from the same hemilineage. We found that *presynapses* of a hemilineage-temporal cohort have greater shared connectivity than observed for unrelated neurons, temporal cohorts, or hemilineages (*Figure 8C*). In contrast, *postsynapses* of a hemilineage-temporal cohort have greater shared connectivity than observed for unrelated neurons and temporal cohorts, but not for hemilineages (*Figure 8D*). Moreover, pairwise comparisons revealed greater hemilineage-temporal cohort interconnectivity than hemilineages alone (*Figure 8E*, top). Furthermore, we detected greater connectivity between neurons within an individual hemilineage-temporal cohort than within an individual hemilineage (*Figure 8E*, bottom). These results suggest that within each hemilineage temporal identity confers an added level of connectivity compared to proximity or hemilineage alone. In conclusion, we propose that neuroblast lineage, hemilineage, and temporal identity function combinatorially to refine neurite projections, synapse localization, and connectivity (*Figure 8F*).

## Hemilineage-temporal cohorts and circuit formation: the Eve proprioceptive circuit

We previously identified a proprioceptive motor circuit containing Eve lateral (EL) interneurons that is required to maintain left/right symmetry of body wall muscle contractions (*Heckscher et al., 2015*; *Figure 9A*). There are three Even-skipped (Eve)+ interneurons at the core of the circuit, receiving strong input from the proprioceptive sensory neurons and three local interneurons (Jaam1-3), and strong output to three premotor neurons (Saaghi1-3) (*Heckscher et al., 2015*). Recent work has shown that the three Eve+ interneurons at the core of the circuit are from a single hemilineage-temporal cohort: late-born neurons from the NB3-3 hemilineage (*Wreden et al., 2017*). This has led to speculation that the input Jaam neurons and the output Saaghi neurons may also be hemilineage-temporal cohorts (*Meng and Heckscher, 2021*). Fortunately, the Eve+, Jaam, and Saaghi neurons are all contained within our data set, allowing us to test this hypothesis. We confirm the results of Wreden et al., showing that the three Eve+ neurons (A08e1-3) are in a single late-born hemilineage-temporal cohort (*Figure 9B*; *Supplementary file 1*). In addition, we find that the Jaam neurons are all derived from a single hemilineage-temporal cohort: late-born neuron in the NB5-2 ventral (Notch$^{OFF}$) hemilineage (*Figure 9B*; *Supplementary file 1*). Similarly, all three Saaghi neurons are derived from a single hemilineage-temporal cohort: late-born neuron in the NB5-2 dorsal (Notch$^{ON}$)

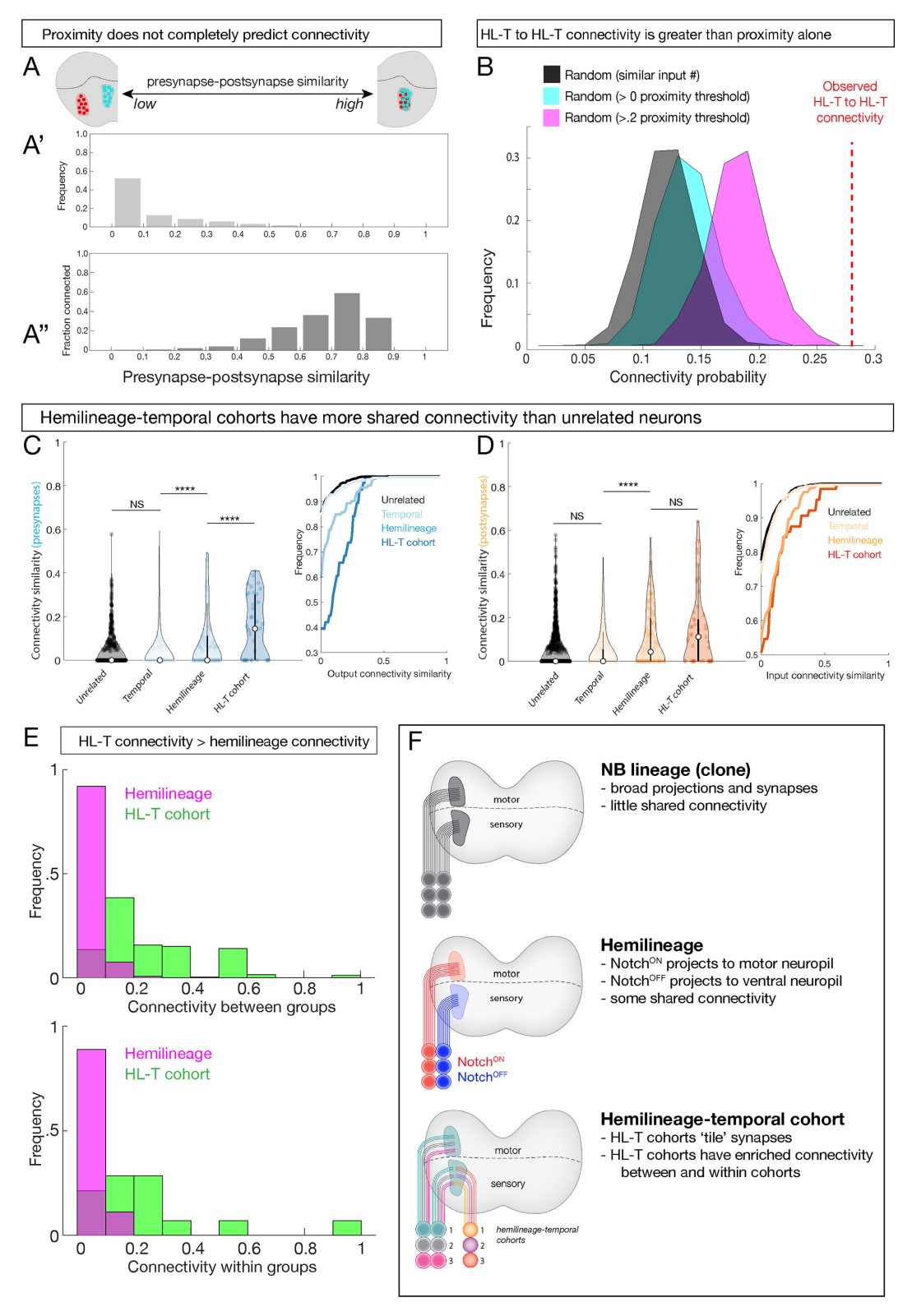

**Figure 8.** Hemilineage-temporal (HL-T) cohorts have greater shared synaptic connectivity than unrelated neurons or hemilineages. (**A-A''**) Axon-dendrite proximity alone does not predict connectivity. The distribution of presynaptic/postsynaptic overlap for all neurons analyzed; higher 'similarity' = smaller average 3D distance between presynapses and postsynapses. (Top) Most neuronal pairs have little overlap (0–0.1 score) in presynapse/postsynapse position. (Bottom) Presynaptic and postsynaptic neurons have increasing connectivity probability as the distance between

*Figure 8 continued on next page*

*Figure 8 continued*

presynapses and postsynapses decreases, but even neurons with the most presynapse/postsynapse overlap often fail to be connected (e.g., 0.7–0.9 similarity are <0.6 connectivity). (**B**) HL-T to HL-T connectivity occurs more frequently than can be explained by proximity alone. Red dashed line indicates observed frequency that a pair of neurons in a HL-T cohort connect to one or more neurons in another HL-T cohort. Two synapse threshold used. Colored distributions represent data shuffled on the basis of proximity, while the black distribution is data shuffled on the basis of input probability (see Materials and methods). (**C, D**) HL-T cohorts have common synaptic partners. Left: violin plots; median: white circle. Right: cumulative plots. (**C**) HL-T cohort *presynapses* have greater shared connectivity than observed for unrelated neurons, temporal cohorts, or hemilineages. (**D**) In contrast, HL-T cohort *postsynapses* have greater shared connectivity than observed for unrelated neurons, temporal cohorts, but not for hemilineages. Connectivity similarity is equivalent to one minus the cosine distance between the presynapses (**C**, blue) or postsynapses (**D**, orange) vectors of the binarized connectivity matrix. ****$p<0.0001$ in a Mann–Whitney test. (**E**) Top: there is greater connectivity between pairs of HL-T cohorts (green) than between pairs of hemilineages (magenta). Bottom: there is greater connectivity within a single HL-T cohort (green) than within a single hemilineage (magenta). (**F**) Summary.

hemilineage (*Figure 9B*; *Supplementary file 1*). Thus, the Jaam-EL-Saaghi proprioceptive circuit is assembled from three distinct hemilineage-temporal cohorts (*Figure 9C*). We propose that other motor circuits may also be assembled by preferential connectivity between distinct hemilineage-temporal cohorts.

## Discussion

Here, we determine the relationship between developmental mechanisms (spatial, temporal, and hemilineage identity) and circuit assembly mechanisms (projections, synapse localization, and connectivity). To do this, we map both developmental and circuit features for 160 neuronal progeny of 14 neuroblast lineages in a serial section TEM reconstruction – this allows us to characterize neurons that share a developmental feature at single synapse resolution. It is important to note that we chose the seven neuroblasts in this study based on successful clone generation and availability of single neuroblast Gal4 lines, and thus there should be no bias towards a particular pattern of neurite projections, synapse localization, or connectivity. Our results show that individual neuroblast lineages have unique but broad axon and dendrite projections to both motor and sensory neuropil; hemilineages restrict projections and synapse localization to either motor or sensory neuropil; and distinct temporal identities within hemilineages provide additional specificity in synapse localization and connectivity. Thus, all three developmental mechanisms act combinatorially to progressively refine neurite projections, synapse localization, and connectivity (*Figure 8F*).

In mammals, clonally related neurons often have a similar location (*Fekete et al., 1994*; *Mihalas and Hevner, 2018*), morphology (*Mihalas and Hevner, 2018*; *Wong and Rapaport, 2009*), and connectivity (*Yu et al., 2009*). In contrast, we found that clonally related neurons project widely in the neuropil, to both sensory and motor domains, and thus lack shared morphology. Perhaps as brain size expands to contain an increasing number of progenitors, each clone takes on a more uniform structure and function. Yet the observation that each neuroblast clone had highly stereotyped projections suggests that neuroblast identity (determined by the spatial position of the neuroblast) determines neuroblast-specific projection patterns. Testing this functionally would require manipulating spatial patterning cues to duplicate a neuroblast and assay both duplicate lineages for similar projections and connectivity.

We found that hemilineages produce sensory and motor processing units via a Notch-dependent mechanism. Pioneering work on *Drosophila* third instar larval neuroblast lineages showed that each neuroblast lineage is composed of two hemilineages with different projection patterns and neurotransmitter expression (*Harris et al., 2015*; *Lacin and Truman, 2016*; *Truman et al., 2010*). We extend these studies to embryonic neuroblasts and show that Notch signaling determines motor versus sensory neuropil projections in all lineages examined. Surprisingly, the Notch[ON] hemilineage always projected to the dorsal/motor neuropil, whereas the Notch[OFF] hemilineage always projected to the ventral/sensory neuropil. The relationship between the Notch[ON] hemilineage projecting to the motor neuropil may be a common feature of all 30 segmental neuroblasts or it could be that the Notch[ON]/Notch[OFF] provides a switch to allow each hemilineage to respond differently to dorsoventral guidance cues, with some projecting dorsally and some projecting ventrally. Analysis of additional neuroblast lineages will resolve this question. Another point to consider is the potential role of

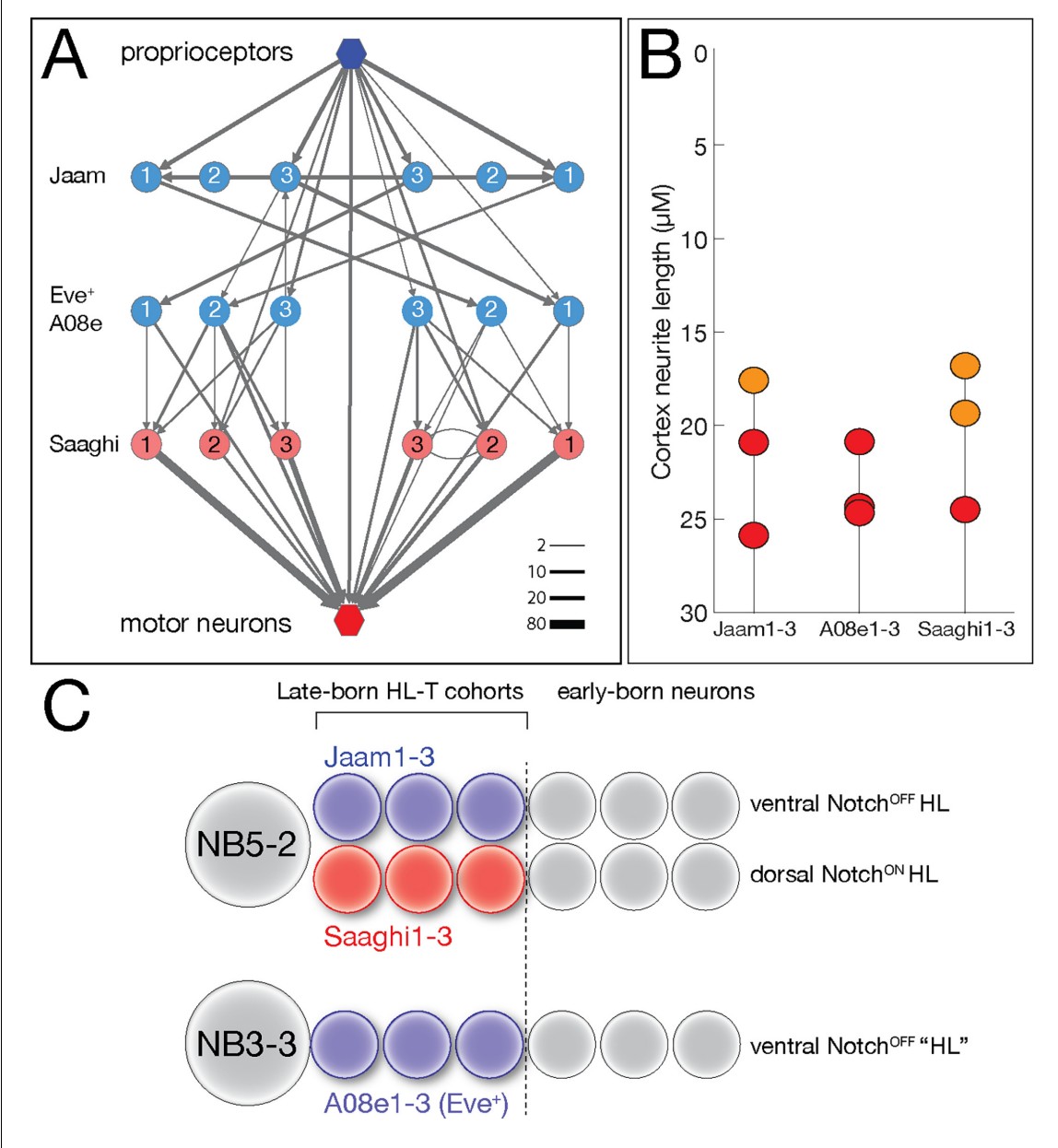

**Figure 9.** Hemilineage-temporal cohorts assemble a proprioceptive circuit. (**A**) The Eve-lateral (EL) proprioceptive circuit (*Heckscher et al., 2015*), including hemilineage-temporal cohort membership. MNs: motor neurons. (**B**) Average cortex neuron length for the nine interneurons shown in (**A**) in the left and right hemisegments; all are in the late-born groups 3 and 4, colored orange and red, respectively (*Supplementary file 1*). (**C**) Developmental origin of the Eve+ proprioceptive circuit. NB5-2 generates a Notch<sup>OFF</sup> hemilineage including late-born Jaam1-3 neurons, and a Notch<sup>ON</sup> hemilineage including the late-born Saaghi1-3 neurons. NB3-3 undergoes a type 0 division pattern where the neuroblast progeny do not divide and remain Notch<sup>OFF</sup> (*Baumgardt et al., 2014*; *Wreden et al., 2017*), effectively creating a Notch<sup>OFF</sup>'hemilineage' that includes the late-born A08e1-3 Eve+ neurons. Thus, the proprioceptive circuit shown in (**A**) comprises three interconnected hemilineage-temporal cohorts.

Notch in post-mitotic neurons (*Crowner et al., 2003*) as our experiments generated Notch<sup>intra</sup> mis-expression in both newborn sibling neurons as well as mature post-mitotic neurons. Future work manipulating Notch levels specifically in mature post-mitotic neurons undergoing process outgrowth will be needed to identify the role of Notch in mature neurons, if any.

Elegant work has identified neuropil gradients of Slit and Netrin along the mediolateral axis (*Zlatic et al., 2009*), Semaphorins along the dorsoventral axis (*Zlatic et al., 2009*), and Wnt5 along the anteroposterior axis (*Yoshikawa et al., 2003*). Our finding that neurons in a hemilineage project

to a common region of the neuropil strongly suggests that all neurons within a hemilineage respond in the same way to these global pathfinding cues. Conversely, our finding that neurons in different hemilineages target distinct regions of the neuropil suggests that each hemilineage expresses a different palette of guidance receptors, which enable them to respond differentially to the same global cues. For example, neurons in ventral hemilineages may express Plexin receptors to repel them from high Semaphorins in the dorsal neuropil.

Hemilineages have not been well described in vertebrate neurogenesis. Notch signaling within the Vsx1 + V2 progenitor lineage generates Notch$^{OFF}$ V2a excitatory interneurons and Notch$^{ON}$ V2b inhibitory interneurons, which may be distinct hemilineages (*Del Barrio et al., 2007*; *Francius et al., 2016*; *Peng et al., 2007*; *Skaggs et al., 2011*). Interestingly, both V2a and V2b putative hemilineages contain molecularly distinct subclasses (*Harris et al., 2019*); our work raises the possibility that these subtypes arise from temporal patterning within the V2 lineage. In addition, Notch$^{ON}$/Notch$^{OFF}$ hemilineages may exist in the pineal photoreceptor lineage, where Notch$^{ON}$ and Notch$^{OFF}$ populations specify cell-type identity (*Cau et al., 2019*).

Only recently have the role of hemilineages been tested for their functional properties. In adults, activation of each larval hemilineage from NB5-2 showed similar behavioral output, whereas each hemilineage from NB6-1 elicited different behaviors (*Harris et al., 2015*). Our previous work showed that the Eve+, Saaghi, and Jaam neurons are part of a proprioceptive circuit (*Heckscher et al., 2015*); here, we show that each class of neurons represents a hemilineage-temporal cohort. Note that the Jaam neurons process sensory input and are in a Notch$^{OFF}$ hemilineage, supporting our conclusion that Notch$^{OFF}$ hemilineages are devoted to sensory processing; the Saaghi premotor neurons are in a Notch$^{ON}$ hemilineage consistent with their role in motor processing. Interestingly, both input and output neurons in this circuit arise from a common progenitor (NB5-2), which may generate late-born Jaam/Saaghi sibling neurons (*Figure 9B*). In the future, it would be interesting to determine if other sibling hemilineages are in a common circuit to generate a specific behavior.

Our hemilineage results have several implications. First, our results reveal that sensory and motor processing components of the neuropil are being built in parallel, with one half of every GMC division contributing to either sensory or motor networks. This would be an efficient mechanism to maintain sensory/motor balance as lineage lengths are modified over evolutionary time. Second, our results suggest that looking for molecular or morphological similarities in full neuroblast clones may be misleading due to the full neuroblast clone comprising two different hemilineages. For example, performing bulk RNAseq on all neurons in a neuroblast lineage is unlikely to reveal key regulators of pathfinding or synaptic connectivity due to the mixture of disparate neurons from the two hemilineages.

We used the cortex neurite length of neurons as a proxy for birth-order and shared temporal identity. We feel this is a good approximation, but it clearly does not precisely identify neurons born during each of the Hb, Kr, Pdm, Cas temporal transcription factor windows. Nevertheless, we had sufficient resolution to observe that neurons with the same temporal identity clustered their pre- or postsynapses, rather than localizing them uniformly through the hemilineage neuropil domain (*Figure 6G*). Interestingly, the three-dimensional location of each hemilineage temporal cohort synaptic cluster is identical on the left and right side of A1 (data not shown), ruling out the mechanism of stochastic self-avoidance (*Zipursky and Grueber, 2013*). Other possible mechanisms include hemilineage-temporal cohorts expressing different levels of the presynapse spacing cue Sequoia (*Kulkarni et al., 2016*; *Petrovic and Hummel, 2008*) or hemilineage-temporal cohorts exhibiting different responses to global patterning cues. Testing the function of temporal identity factors in synaptic tiling will require hemilineage-specific alteration of temporal identity, followed by assaying synapse localization within the neuropil.

Our results strongly suggest that hemilineage identity and temporal identity act combinatorially to allow small pools of neurons to target pre- and postsynapses to highly precise regions of the neuropil, thereby restricting synaptic partner choice. Yet precise neuropil targeting is not sufficient to explain connectivity as many similarly positioned axons and dendrites fail to form connections (*Figure 8C*). We favor the model that hemilineages direct gross neurite targeting to motor or sensory neuropil, whereas temporal identity acts combinatorially with each hemilineage to direct more precise neurite targeting and synaptic connectivity. Thus, the same temporal cue (e.g., Hb) could promote targeting of one pool of neurons in one hemilineage and another pool of neurons in an

adjacent hemilineage. This limits the number of regulatory mechanisms needed to generate precise neuropil targeting and connectivity for all ~600 neurons in a segment of the larval CNS.

In conclusion, we demonstrate how developmental information can be integrated with connectomic data. We show that lineage information, hemilineage identity, and temporal identity can all be accurately predicted using morphological features (e.g., number of fascicles entering the neuropil for neuroblast clones and radial position for temporal cohorts). This both greatly accelerates the ability to identify neurons in a large EM volume as well as sets up a framework in which to study development using data typically intended for studying connectivity and function. We have used this framework to relate developmental mechanism to neuronal projections, synapse localization, and connectivity. We find that lineage, hemilineage, and temporal identity act sequentially to progressively refine neuronal projections, synapse localization, and connectivity, and our data supports a model where hemilineage-temporal cohorts are units of connectivity for assembling motor circuits.

# Materials and methods

## Key resources table

| Reagent type (species) or resource | Designation | Source or reference | Identifiers | Additional information |
|---|---|---|---|---|
| Genetic reagent (*Drosophila melanogaster*) | R16A05[AD] R28H10[DBD] | *Lacin et al., 2019* | RRID:BDSC_70900 RRID:BDSC_69496 | NB1-2 split Gal4 |
| Genetic reagent (*Drosophila melanogaster*) | R70D06[AD] R28H10[DBD] | *Lacin et al., 2019* | RRID:BDSC_69496 RRID:BDSC_70900 | NB2-1 split Gal4 |
| Genetic reagent (*Drosophila melanogaster*) | Ac[AD] Gsb[DBD], 25A05[kz] | *Seroka and Doe, 2019* | RRID:BDSC_70983 | NB7-1 split Gal4 |
| Genetic reagent (*Drosophila melanogaster*) | R19B03[AD] R18F07[DBD] | *Lacin et al., 2019* | RRID:BDSC_70579 RRID:BDSC_70047 | NB7-4 split Gal4 |
| Genetic reagent (*Drosophila melanogaster*) | castor-gal4 | Gift from Technau lab | | Late-born neuron marker |
| Genetic reagent (*Drosophila melanogaster*) | hsFlp.G5.PEST.Opt | BDSC | RRID:BDSC_77140 | Heat-inducible Flp recombinase |
| Genetic reagent (*Drosophila melanogaster*) | 26XLexAop2-mCD8::GFP | BDSC | RRID:BDSC_32207 | LexA reporter |
| Genetic reagent (*Drosophila melanogaster*) | 13XLexAop2-IVS-myr::GFP | BDSC | RRID:BDSC_32210 | LexA reporter |
| Genetic reagent (*Drosophila melanogaster*) | dpn(FRT.stop)LexA.p65 | BDSC | RRID:BDSC_56162 | Used with hsFlp and lexAop-GFP to visualize clones in single neuroblasts |
| Genetic reagent (*Drosophila melanogaster*) | 13XLexAop2-IVS-myr::GFP | BDSC | RRID:BDSC_32210 | LexA reporter |

*Continued on next page*

*Continued*

| Reagent type (species) or resource | Designation | Source or reference | Identifiers | Additional information |
|---|---|---|---|---|
| Genetic reagent (*Drosophila melanogaster*) | w[1118] P{y[+t7.7] w[+mC]=R57 C10-FLPG5.PEST} attP18; P{y[+t7.7] w[+mC]=10xUAS(FRT.stop)myr::smGdP-OLLAS} attP2 PBac{y[+mDint2] w[+mC]=10xUAS(FRT.stop)myr::smGdP-HA}VK00005 P{10xUAS(FRT.stop) myr::sm GdP-V5-THS-10xUAS (FRT.stop)myr::smGdP-FLAG}su(Hw)attP1 | BDSC | RRID:BDSC_64091 | Multicolor Flp Out stock (*Nern et al., 2015*) |
| Genetic reagent (*Drosophila melanogaster*) | Sco/CyO; Dr/TM3,Sb | BDSC | RRID:BDSC_34516 | |
| Genetic reagent (*Drosophila melanogaster*) | LexA-T2A-Hb | This work | | Endogenous hb locus CRISPR engineered to place LexAp65-T2A upstream and in frame with the first hb ORF |
| Genetic reagent (*Drosophila melanogaster*) | hey:T2A:FLP, UAS-myr:GFP, UAS-FRT-stop-FRT-myr:sfGdP:HA | This work | | Labels Hey+ (Notch$^{ON}$) neurons within a Gal4+ neuronal population |
| Antibody, polyclonal | Rabbit anti-GFP A-11122 | ThermoFisher, Waltham, MA | RRID:AB_221569 | 1:500 |
| Antibody, polyclonal | Chicken anti-GFP | Abcam, Eugene, OR | RRID:BDSC_13970 | 1:1000 |
| Antibody, polyclonal | Camelid sdAB direct labeled with AbberiorStar635P 'Fluo Tag-Q anti-GFP' #N0301 | NanoTab Biotech., Gottingen, Germany | | 1:1000 |
| Antibody, polyclonal | Rabbit anti-mCherry NBP2-25157 | Novus, Littleton, CO | RRID:AB_2753204 | 1:1000 |
| Antibody, polyclonal | Alexa Fluor 488-conjugated rabbit anti-GFP NBP1-69969 | ThermoFisher (Eugene, OR) | RRID:AB_221477 | 1:1000 |
| Antibody, monoclonal | Mouse anti-FasII 1D4 | DSHB (Iowa City, IA) | RRID:AB_528235 | 1:100 |
| Antibody, monoclonal | Mouse anti-HA(6E2) #2350 | Cell Signaling, Danvers, MA | RRID:AB_491023 | 1:200 |
| Antibody, polyclonal | Rabbit anti-V5 Dylite 549 #600-442-378 | Rockland, Atlanta, GA | RRID:AB_1961802 | 1:400 |
| Antibody, polyclonal | Rabbit anti-FLAG Dylite488 # 600-441-383 | Rockland, Atlanta, GA | RRID:AB_1961508 | 1:200 |
| Antibody, monoclonal | Mouse anti-Engrailed 4D9 | DSHB (Iowa City, IA) | RRID:AB_528224 | 1:100 |
| Antibody, polyclonal | Rabbit anti-Hb | Doe lab | | 1:400 |
| Antibody, polyclonal | Alexa Fluor 405 Phalloidin | ThermoFisher (Eugene, OR) | | 1:40 |
| Antibody, polyclonal | Secondary antibodies | ThermoFisher (Eugene, OR) | | 1:400 |

*Continued on next page*

*Continued*

| Reagent type (species) or resource | Designation | Source or reference | Identifiers | Additional information |
|---|---|---|---|---|
| Sequence-based reagent | pHD-DsRed | Addgene | RRID:Addgene_51434 | |
| Sequence-based reagent | pCFD5 | Addgene | RRID:Addgene_73914 | |

## Transgenic fly stocks

Transgenic lines were made by BestGene (Chino Hills, CA) or Genetivision (Houston, TX).

## NB clone generation and lineage identification

The NB clones were generated with the following flies: *hs-Flp.G5.PEST.Opt, dpn(FRT.stop)LexA.p65, 26XLexAop2-mCD8::GFP.* The embryos were collected in 25℃ for 3 hr and then incubated in 25℃ for another 3 hr. The aged embryos were then submerged in 32℃ water bath for 5 min heat shock and then incubated in 25℃ until larvae hatched. The CNS of newly hatched larvae was dissected and mounted as previously described (*Clark et al., 2016*; *Heckscher et al., 2014*; *Syed et al., 2017*). The neuropil was stained with mouse anti-engrailed (RRID:AB_528224) (DSHB, 4D9) and Alexa Fluor 647 Phalloidin (ThermoFisher) by following the manufacturer's protocol. The images were collected with Zeiss710 and processed with Imaris.

Lineages were identified in the EM volume by finding neurons with morphologies that matched the clonal morphology and then identifying their neuropil entry point. We then examined every neuron that entered the neuropil in the same fascicle. In most cases, every neuron in the fascicle had a morphology that matched the clonal morphology. In a small number of cases, the fascicles diverged slightly before the neuropil entry point. We verified the number of neurons by looking at fasciculating cell populations from at least two hemisegments (A1L and A1R). In some cases, we were able to identify a stereotyped number of cells across as many as four hemisegments, suggesting that fasciculation is stereotyped and reliable.

## Hb+ single-cell clone generation

Hb+ single-cell clones were generated with the following flies: *hs-Flp.G5.PEST.Opt*, *13XLexAop2-IVS-myr::GFP*, and *LexAp65-T2A-hb* (see below). The embryos were collected for 7 hr in 25℃, submerged in 32℃ water bath for 10 min heat shock, and then followed the protocol as described above.

*LexAp65-T2A-hb* was generated by in-frame fusion of (*FRT.stop)::LexA.P65::T2A* to the N-terminus of the hb open reading frame with CRISPR-Cas9 gene editing. The ds-DNA donor vector for homology-directed repair was composed of left homologous arm (1000 bp), LexA.P65 (*Pfeiffer et al., 2008*), T2A (*Nern et al., 2015*), and the right homologous arm (1000 bp); the fragments were amplified with PCR and then assembled in pHD-DsRed (RRID:Addgene_51434) with NEBuilder (New England BioLabs). The gRNAs were generated from the vector pCFD5 (RRID:Addgene_73914) (*Port et al., 2014*) containing target sequence TGCATCTTGGCGGCTCTAGA and ACTACGAGCAGCACAACGCC. The ds-DNA donor vectors and gRNA vectors were co-injected into *yw;nos-Cas9* (*Kondo and Ueda, 2013*) flies by BestGene. The selection marker 3xP3-DsRed was then removed in transgenic flies by *hs-Cre*.

## Immunostaining and imaging

Standard confocal microscopy, immunocytochemistry, and MCFO methods were performed as previously described for larvae (*Clark et al., 2016*; *Heckscher et al., 2014*; *Syed et al., 2017*) or adults (*Nern et al., 2015*; *Pfeiffer et al., 2008*). Secondary antibodies were from Jackson Immunoresearch (West Grove, PA) and used according to the manufacturer's instructions. Confocal image stacks were acquired on Zeiss 700, 710, or 800 microscopes. Images were processed in Fiji (https://imagej.net/Fiji), Adobe Photoshop (Adobe, San Jose, CA), and Adobe Illustrator (Adobe). When adjustments to brightness and contrast were needed, they were applied to the entire image uniformly. Mosaic images to show different focal planes were assembled in Fiji or Photoshop.

## Morphological analysis of lineages

Morphological analysis was done using NBLAST and the NAT package (*Costa et al., 2016*), and analysis and figure generation were done using R. Neurons were preprocessed by pruning the most distal twigs (Strahler order 4), converting neurons to dot-props, and running an all-by-all NBLAST. For individual lineages, clusters were set using a cutoff of 3.0. In the case of NB2-1, where nearly every neuron shares a very similar morphology, we first confirmed the presence of a hemilineage using anti-Hey staining. After confirmation of a hemilineage, we next removed A02o and A02l since we could not find any clones that contained either an anterior projection (A02o) or a second contra-lateral projection (A02l). We reasoned that the hemilineages would represent the next largest morphological division (*Figure 2—figure supplement 1*).

## Synaptic distributions and density analysis

Synapse distribution plots and density contours were generated using MATLAB. Neuron synaptic and skeleton information was imported to MATLAB using pymaid (*Schlegel et al., 2016*). Cross-sectional synapse distribution plots were made by taking all synapse positions between the T3 and A2 segments as positional information becomes lost due to changes in brain shape beyond these bounds. Synapse distribution plots are 1D kernel density estimates. Sensory and motor density maps were made by taking the synapse positions of all sensory neurons entering the A1 nerve, and all motor neurons exiting the A1 nerve as well as all neurons with at least three synapses connected to one of these neurons. For sensory and motor maps as well as individual hemilineages, density plots are 2D kernel density estimates of all synapse positions across the neuropil. A cutoff of 60% was used to set the outermost contour. For lineage maps (*Figure 4A*), we used 80% as a cutoff. Polyadic synapses were counted as many times as they have targets. For synapse distribution plots, polyadic synapses are scaled by their number of targets.

## Temporal cohort assignment

Cortex neurite length was calculated by converting the skeletonized neuronal arbor into a directed graph away from the soma and performing a depth-first-search of all vertices. The neuropil borders were defined by a previously created neuropil volume object (*Costa et al., 2016*). The neuropil entry point was defined as the first vertex within the neuropil volume object. Cortex neurite length was then the path length between the soma and the neuropil entry point. Neurons were binned into four groups defined by the positions of identified Hb+ and Cas+ cells. Early-born cells were defined as neurons with a cortex length <1 standard deviation above the mean Hb+ neurite length. The next group had cortex neurite lengths ≤1 standard deviation below the mean Cas+ neurite length. The final two groups were split at the mean Cas+ neurite length.

## Synapse similarity measurements

Synapse similarity was calculated as described previously (*Schlegel et al., 2016*):

$$f(is,jk) = e^{\frac{-d_{sk}^2}{2\sigma^2}} e^{-\frac{|n_{is}-n_{jk}|}{n_{is}+n_{jk}}}$$

where $f(is,jk)$ is the mean synapse similarity between all synapses of neuron $i$ and neuron $j$. $d_{sk}$ is the Euclidean distance between synapses $s$ and $k$ such that synapse $k$ is the closest synapse of neuron $j$ to synapse $s$ of neuron $i$. $\sigma$ is a bandwidth term that determines what is considered close. $n_{is}$ and $n_{jk}$ are the fraction of synapses for neuron $i$ and neuron $j$ that are within $\omega$ of synapse $s$ and synapse $k$, respectively. We used parameters $\omega = \sigma = 2000$ nm. For presynaptic or postsynaptic comparisons, the score for a given pair of neurons was the average similarity between the left and right sides. In order to calculate pre/post overlap, we applied the same measure, instead using the presynapses of neuron $i$ and the postsynapses of neuron $j$.

## Connectivity similarity measurement

Connectivity similarity was calculated as one minus the cosine distance between either the row vectors (outputs) or column vectors (inputs) of the binarized connectivity matrix for all neurons. For models of unrelated neuron connectivity, we used two methods to determine the individual connection probabilities. First, we used a standard degree-based method in which the number of inputs for

a given neuron was divided by the total number of inputs received by that neuron. Second, we used the pre-postsynapse overlap scores. For a given neuron, its probability of connecting to another neuron was equal to the overlap score divided by the sum of all overlap scores for that neuron. Pairs with a score of 0 had a 0 probability of connecting. Thresholds were done by setting all values below the threshold to zero when determining connection probability.

## Electron microscopy and CATMAID

We reconstructed neurons in CATMAID as previously described (*Carreira-Rosario et al., 2018*; *Heckscher et al., 2015*; *Ohyama et al., 2015*).

## Figures

Figures were generated using MATLAB, R, CATMAID, and Fiji, and edited in either Illustrator or Photoshop (Adobe).

## Statistical analysis

Statistical significance is denoted by asterisks: ****$p < 0.0001$; ***$p < 0.001$; **$p < 0.01$; *$p < 0.05$; n.s., not significant. All statistical analyses were done in MATLAB. When comparing two groups of quantitative data, an unpaired *t*-test was performed if data was normally distributed (determined using a one-sample Kolmogorov–Smirnov test) and Wilcoxon rank-sum test if the data was not normally distributed. Two-sample Kolmogorov–Smirnov tests were used on empirical distributions. Linear models were generated in MATLAB using lmfit.

## Data availability

All data are publicly available from https://github.com/bjm5164/Mark2020_larval_development.

# Acknowledgements

We thank Haluk Lacin for unpublished fly lines. We thank Todd Laverty, Gerry Rubin, and Gerd Technau for fly stocks; Luis Sullivan, Emily Sales, John Tuthill, and Tim Warren for comments on the manuscript; Avinash Khandelwal and Laura Herren for annotating neurons; Keiko Hirono for generating transgenic constructs; and Keiko Hirono, Rita Yazejian, Dalton Lee, Cooper Doe, and Casey Doe for confocal imaging. Stocks were obtained from the Bloomington *Drosophila* Stock Center (NIH P40OD018537). Funding was provided by HHMI (CQD, BM, LM, AAZ), NIH HD27056 (CQD), and NIH T32HD007348-24 (BM).

# Additional information

### Competing interests

Chris Q Doe: Reviewing editor, *eLife*. Albert Cardona: Reviewing editor, *eLife*. The other authors declare that no competing interests exist.

### Funding

| Funder | Grant reference number | Author |
| --- | --- | --- |
| National Institutes of Health | HD27056 | Brandon Mark<br>Sen-Lin Lai<br>Aref Arzan Zarin<br>Laurina Manning<br>Heather Q Pollington<br>Chris Q Doe |
| HHMI | | Brandon Mark<br>Sen-Lin Lai<br>Aref Arzan Zarin<br>Laurina Manning<br>Heather Q Pollington<br>Chris Q Doe |

| | | |
|---|---|---|
| HHMI | | Albert Cardona<br>James W Truman |
| National Institutes of Health | T32HD007348-24 | Brandon Mark |
| National Institutes of Health | 5T32GM007413-43 | Heather Q Pollington |

The funders had no role in study design, data collection and interpretation, or the decision to submit the work for publication.

## Author contributions

Brandon Mark, Conceptualization, Resources, Data curation, Software, Formal analysis, Validation, Investigation, Visualization, Methodology, Writing - original draft, Writing - review and editing; Sen-Lin Lai, Investigation, Methodology; Aref Arzan Zarin, Formal analysis, Investigation, Methodology; Laurina Manning, Investigation; Heather Q Pollington, Formal analysis, Validation, Investigation, Methodology; Ashok Litwin-Kumar, Formal analysis, Investigation; Albert Cardona, Resources; James W Truman, Resources, Data curation, Formal analysis, Validation; Chris Q Doe, Conceptualization, Resources, Formal analysis, Supervision, Funding acquisition, Validation, Writing - original draft, Project administration, Writing - review and editing

## Author ORCIDs

Sen-Lin Lai http://orcid.org/0000-0002-7531-283X
Aref Arzan Zarin http://orcid.org/0000-0003-0484-3622
Ashok Litwin-Kumar http://orcid.org/0000-0003-2422-6576
Albert Cardona http://orcid.org/0000-0003-4941-6536
James W Truman http://orcid.org/0000-0002-9209-5435
Chris Q Doe https://orcid.org/0000-0001-5980-8029

## Decision letter and Author response

Decision letter https://doi.org/10.7554/eLife.67510.sa1
Author response https://doi.org/10.7554/eLife.67510.sa2

# Additional files

## Supplementary files

• Supplementary file 1. All the neurons described in this paper. Neurons are annotated for names/synonyms, parental neuroblast, hemilineage, and radial position.

• Transparent reporting form

## Data availability

All data are publicly available from https://github.com/bjm5164/Mark2020_larval_development (copy archived at https://archive.softwareheritage.org/swh:1:rev:43e0a22c5381427aa6670c55ec4de76f5ad39568).

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
