## [Decision Letter]

**Acceptance summary:**

This paper will be of high interest to a broad audience of developmental neurobiologists, since it addresses how specification mechanisms in neural stem cells impact circuit connectivity of their neuronal progeny. By using TEM and light microscopy of the larval *Drosophila* CNS, as well as leveraging the deep understanding of lineage of embryonic neuroblasts, the work establishes the cellular rules contributing to wiring specificity, unveiling a role for Notch in the concurrent building of sensory and motor circuits in each lineage.

**Decision letter after peer review:**

Thank you for submitting your article "A developmental framework linking neurogenesis and circuit formation in the *Drosophila* CNS" for consideration by *eLife*. Your article has been reviewed by 3 peer reviewers, and the evaluation has been overseen by a Reviewing Editor and Ronald Calabrese as the Senior Editor. The following individuals involved in review of your submission have agreed to reveal their identity: Isabel Holguera (Reviewer #2); Bassem A Hassan (Reviewer #3).

Essential revisions:

The three reviewers are quite enthusiastic about the quality of the work and the potential implications. However, they feel that you need to tone down very significantly your conclusions: You should provide in the text less categorical interpretations of your results and discuss more broadly the different possible alternatives that explain the results you present.

– Although it would have ben nice to test the role of Notch in hemilineages using LoF and in more than one lineage (perhaps you already have these data?), it would be sufficient to clearly state that while your data support a role for Notch in organizing targeting, it remains to be determined whether this is through hemilineage fate specification and/or additional regulation of neuronal targeting at later developmental stages. Discuss the possibility that hemilineage temporal cohorts correlate with developmental proximity without drawing conclusions.

– Tone down your statements on the exclusivity of target space occupation by different cohorts because it is obvious from your data that there is quite a bit of mixing. You should also explain the meaning of the 2 synapses or less rule.

*Reviewer #1 (Recommendations for the authors):*

This is a remarkable study. The authors might consider how to make the work more accessible to readers interested in related issues in the mammalian brain. My concern is that the detailed analysis of so many different neurons and lineages will be a blur for readers unfamiliar with the system.

*Reviewer #2 (Recommendations for the authors):*

Suggestions to improve the paper:

Line 116. "We note that some of the earliest born neurons are not included either because their cell bodies are in contact with the neuropil and they do not fasciculate with clonal fascicles, precluding assignment to a specific neuroblast lineage, or they do not maintain marker expression at larval hatching. The morphology and function of the earliest born neurons will be described elsewhere."

This paragraph is not sufficiently developed in the text. By looking at Figure 1E, NB3-3, NB4-1 and NB7-4 seem to be the ones that do not have neuronal somas localized close to the neuropil and hence the early born neurons the authors are referring to. Could you indicate this in Figure 1C to make this claim easier to understand? Moreover, you could refer to Figure 5I, where the authors do not show early born neurons for these lineages.

Line 139: "In the case of NB3-3, where only a single class of neurons was observed, all neurons were ventral-projecting (Figure 2I)."

The authors claim only a single morphological class is made by this lineage, but the NBLAST clustering in the Figure has two different branches, in the same manner as the NBLAST clustering in the rest of the panels, where two different morphological types of neurons are said to be made. Please, clarify this.

Line 141. "We conclude that most lineages do not generate neurons that share common projections, but rather generate two distinct classes of neurons that target either the motor or sensory neuropil."

This statement is confusing: half of the neurons from a lineage share a common projection, either dorsal or ventral, hence lineages generate two group of neurons, which share projections within each group (hemilineage) but not between groups.

Line 329. "In conclusion, we propose that neuroblast lineage, hemilineage, and temporal identity function to progressively refine neurite projections, synapse localization and connectivity (Figure 8F)."

I don't agree with the use of the word "progressively" here: a given neuron is going to acquire its temporal identity and either Notch ON or Notch OFF (hemilineage) identity at the same time (right at birth), so these factors would be acting concurrently.

Line 365. "Thus, all three developmental mechanisms act sequentially to progressively refine neurite projections, synapse localization and connectivity (Figure 8F)."

The same comment as in line 329, regarding the word "sequentially".

Line 373: "Yet the observation that each neuroblast clone had highly stereotyped projections suggests that neuroblast identity (determined by the spatial position and temporal identity of the neuroblast) determines neuroblast-specific projection patterns".

Line 528. "Neurons were binned into 5 groups with 6μm edges to define temporal cohorts."

In the main text 4 temporal cohorts are described, and it is not clear to me from Figure 5I what "6μm edges" mean, since temporal cohorts seem to be comprising approximately 1 to 10, 10 to 15, 15 to 20 and 20 to 30μm groups.

Figure 3 G vs H (NB 1-2): It is not clear from this image that there is an increase in dorsal projections. It just look like ventral ones are lost, but no new dorsal ones have been created.

Figure 4D: NB-1-2 (red) does not look to have presynapses in the ventral part of the neuropil. Why?

Figure 4 CDEF: From these Figures it looks like there are more postsynapses than presynapses. Could it be that postsynapses target more overlapping domains because there is a higher number of postsynapses than presynapses?

Recommendations to improve the presentation and clarity of the Figures:

Figure 1A: indicate the anterior/posterior, medio/lateral and dorso/ventral axes for these neuroblasts.

Figure 1B. To make interpretation of the images easier please denote neuropil and cortex structures. How are the neuropil borders determined? Why is the neuropil region smaller in NB7-4?

Figure 1C. Please indicate with arrows the fascicles entering the neuropil that are shown in Figure 1D.

Figure 1G: What "L-R" means should be written in the figure legend.

Figure 2. Notch ON/Notch OFF is not written in panels F and G.

Supplementary Table 1 would be easier to read and interpret if NB lineages and hemilineages were depicted with colors and also if lines were ordered by temporal cohort (from early to late) instead of by the name of the neuron.

*Reviewer #3 (Recommendations for the authors):*

A. The authors should:

1. Provide a more quantitative assessment of the degree of variation and overlap of neuronal targeting and modify their categorical statements accordingly.

2. Clarify why the "2 synapses or less" observation represents a significant difference between the hemilineage temporal cohorts compared to less related neurons.

B. The authors can choose to either:

1. Perform:

– Spatio-temporally controlled Notch loss and gain of function analyses on an example set of neurons (e.g. in one typical hemilineage);

– Live imaging of how and when neurons invaded a target space and establish synapses

Or:

2. Tone down their conclusions about:

– How Notch activity regulates targeting;

– The extent to which hemilineage temporal cohorts correlate with developmental proximity.

---

## [Author Response]

Reviewer #1 (Recommendations for the authors):This is a remarkable study. The authors might consider how to make the work more accessible to readers interested in related issues in the mammalian brain. My concern is that the detailed analysis of so many different neurons and lineages will be a blur for readers unfamiliar with the system.

Thank you. We agree that it is a complex dataset. To put our work in context and provide links between the neurons in our work and neurons described by others we will deposit our data into the **Virtual Fly Brain** (https://v2.virtualflybrain.org/), which is designed to integrate data on all larval neurons within the brain and VNC.

Reviewer #2 (Recommendations for the authors):Suggestions to improve the paper:Line 116. "We note that some of the earliest born neurons are not included either because their cell bodies are in contact with the neuropil and they do not fasciculate with clonal fascicles, precluding assignment to a specific neuroblast lineage, or they do not maintain marker expression at larval hatching. The morphology and function of the earliest born neurons will be described elsewhere."This paragraph is not sufficiently developed in the text. By looking at Figure 1E, NB3-3, NB4-1 and NB7-4 seem to be the ones that do not have neuronal somas localized close to the neuropil and hence the early born neurons the authors are referring to. Could you indicate this in Figure 1C to make this claim easier to understand? Moreover, you could refer to Figure 5I, where the authors do not show early born neurons for these lineages.

Thanks for these comments. While most lineages in Figure 1C have a gap between the most proximal cell body and the neuropil, where a missing early-born neuron may reside (3-3, 4-1, 5-2, 7-1, 7-4), it is also possible that there is a missing early-born neuron in lineages where the most proximal neuron is close to the neuropil (1-2, 2-1). We add new text on this point “This can sometimes, but not always, lead to a gap between the deepest mapped neuron and the neuropil (Figure 1C).” Regarding Figure 5I, agree it would be best to add text stating that the earliest-born neurons are not represented. We add new text on this point: “We assigned 70 interneurons in our dataset to one of four temporal cohorts (early, mid-early, mid-late and late born) based on radial position (Figure 5I). We note that some of the earliest born neurons are not included (see Methods)”.

Line 139: "In the case of NB3-3, where only a single class of neurons was observed, all neurons were ventral-projecting (Figure 2I)."The authors claim only a single morphological class is made by this lineage, but the NBLAST clustering in the Figure has two different branches, in the same manner as the NBLAST clustering in the rest of the panels, where two different morphological types of neurons are said to be made. Please, clarify this.

We agree this was confusing. The NBLAST dendrograms were thresholded the same across all lineages to show that the first cut almost always separates the lineage into two hemilineages. For NB3-3 this same threshold showed one hemilineage (which has been shown experimentally by the Thor lab). You are right that there is an intriguing pair of subclusters within the single 3-3 hemilineage, and interestingly they map to neurons with early vs late temporal identity. We have clarified the text to say:

"We used NBLAST (Costa et al., 2016) to quantify the morphological similarity of neurons within each neuroblast lineage, and found that most lineages made two distinct classes of neurons, and the largest distinction corresponded to classes of neurons that projected to either the dorsal/motor neuropil or the ventral/sensory neuropil (or made one group of neurons and one group of glia) (Figure 2D-H). Note that NB3-3 had only one cluster at the NBLAST threshold used, and all neurons were ventral-projecting (Figure 2I). These classes were more distinct from one another than from neurons in other lineages (Figure 2—figure supplement 2). We conclude that most lineages generate two distinct classes of neurons that target either the sensory or motor neuropils."

Line 141. "We conclude that most lineages do not generate neurons that share common projections, but rather generate two distinct classes of neurons that target either the motor or sensory neuropil."This statement is confusing: half of the neurons from a lineage share a common projection, either dorsal or ventral, hence lineages generate two group of neurons, which share projections within each group (hemilineage) but not between groups.

We agree this was confusing, and have clarified it: “We conclude that most lineages generate two distinct classes of neurons that target either the sensory or motor neuropils.”

Line 329. "In conclusion, we propose that neuroblast lineage, hemilineage, and temporal identity function to progressively refine neurite projections, synapse localization and connectivity (Figure 8F)."I don't agree with the use of the word "progressively" here: a given neuron is going to acquire its temporal identity and either Notch ON or Notch OFF (hemilineage) identity at the same time (right at birth), so these factors would be acting concurrently.

Yes, good point. We have changed it to “In conclusion, we propose that neuroblast lineage, hemilineage, and temporal identity function combinatorially to refine…”

Line 365. "Thus, all three developmental mechanisms act sequentially to progressively refine neurite projections, synapse localization and connectivity (Figure 8F)."The same comment as in line 329, regarding the word "sequentially".

Yes, we agree and have made the same correction: “Thus, all three developmental mechanisms act combinatorially to progressively refine…”

Line 373: "Yet the observation that each neuroblast clone had highly stereotyped projections suggests that neuroblast identity (determined by the spatial position and temporal identity of the neuroblast) determines neuroblast-specific projection patterns".

Thanks for catching this bad sentence. We have fixed it: “In contrast, we found that clonally-related neurons project widely in the neuropil, to both sensory and motor domains, and thus lack shared morphology.”

Line 528. "Neurons were binned into 5 groups with 6μm edges to define temporal cohorts."In the main text 4 temporal cohorts are described, and it is not clear to me from Figure 5I what "6μm edges" mean, since temporal cohorts seem to be comprising approximately 1 to 10, 10 to 15, 15 to 20 and 20 to 30μm groups.

Thank you for pointing out this error. We have fixed the text: “Neurons were binned into four groups defined by the positions of identified Hb+ and Cas+ cells. Early born cells were defined as neurons with a cortex length less than one standard deviation above the mean Hb+ neurite length. The next group had cortex neurite lengths less than or equal to one standard deviation below the mean Cas+ neurite length. The final two groups were split at the mean Cas+ neurite length.”

Figure 3 G vs H (NB 1-2): It is not clear from this image that there is an increase in dorsal projections. It just look like ventral ones are lost, but no new dorsal ones have been created.

Good point. We have replaced panel H (NB1-2) with a more representative image. We have also added cell counts for wild type and Notch^intra^ for each lineage, showing that cell numbers are not significantly different in the two genotypes (thus Notchintra does not cause loss of ventral projections due to cell death). Thanks.

Figure 4D: NB-1-2 (red) does not look to have presynapses in the ventral part of the neuropil. Why?

Great catch. This is because the NB1-2 ventral hemilineage presynapses are located in the posterior segments and thus are not shown in the A1 cross-sectional view. We have added a plot of the distribution of synapses along the AP axis, which makes this clear, and we add the following to the legend: “Note that NB1-2 ventral hemilineage presynapses (red dots) are located ventrally, but are not shown in the cross-section view due to their position in posterior segments of the VNC.” Thanks for this important comment which has clarified a potentially confusing issue.

Figure 4 CDEF: From these Figures it looks like there are more postsynapses than presynapses. Could it be that postsynapses target more overlapping domains because there is a higher number of postsynapses than presynapses?

It looks that way because fly synapses are polyadic and therefore a single presynaptic site connects with many postsynaptic sites. For clarity on this, each dot on the presynapse distribution plots are scaled in size by the number of postsynaptic sites they are associated with.

Recommendations to improve the presentation and clarity of the Figures:Figure 1A: indicate the anterior/posterior, medio/lateral and dorso/ventral axes for these neuroblasts.

Thanks; we add this to the figure legends: “(A) Three mechanisms specifying neuronal diversity. Neuroblasts characterized here are shown in dark gray and arise from all anteroposterior and mediolateral positions of the neuroectoderm (dorsal view: anterior up, ventral midline at left of panel).”

Figure 1B. To make interpretation of the images easier please denote neuropil and cortex structures. How are the neuropil borders determined? Why is the neuropil region smaller in NB7-4?

Thanks, that was a mistake. We have fixed it.

Figure 1C. Please indicate with arrows the fascicles entering the neuropil that are shown in Figure 1D.

Thanks, we have added the requested arrows.

Figure 1G: What "L-R" means should be written in the figure legend.

We have made the requested change: “L, left; R, right; D, dorsal; V, ventral; A, anterior.:”

Figure 2. Notch ON/Notch OFF is not written in panels F and G.

We are sorry for the confusion. At this point in the paper we have not introduced the Notch ON / Notch OFF analysis. This figure simply shows that using NBLAST we can define two morphological classes of neurons: dorsal projecting and ventral projecting. In the following figure we show that the former is NotchON and the latter is NotchOFF.

Supplementary Table 1 would be easier to read and interpret if NB lineages and hemilineages were depicted with colors and also if lines were ordered by temporal cohort (from early to late) instead of by the name of the neuron.

We agree, and have made the requested changes. Much nicer now.

Line 277: "This allows us to test the hypothesis that hemilineage-temporal cohorts have more shared connectivity other developmental groupings."

Fixed: “This allows us to test the hypothesis that hemilineage-temporal cohorts have more shared connectivity compared to other developmental groupings.”

Reviewer #3 (Recommendations for the authors):A. The authors should:1. Provide a more quantitative assessment of the degree of variation and overlap of neuronal targeting and modify their categorical statements accordingly.

We have toned down our language to eliminate the phrase “striking similarity.” Elegant work from Tzumin Lee and Jim Truman have shown that clones from larval neuroblasts are very similar, and our qualitative findings support this conclusion. Thus, it would be only a minor advance for us to quantify clonal similarity in embryonic neuroblasts. Plus, since the number of neurons in a clone varies slightly, we would have to count neuron numbers per clone and only compare those with identical neuron numbers, which is possible but time-consuming. Then there are the covid restrictions which make it difficult to rapidly generate new clones to increase the number with identical neurons. All in all, we decided that the benefit of answering this question was not worth the cost of performing it, and that other experiments were a higher priority in our limited research time. We have toned down the language to remove the word “striking” in the Introduction.

2. Clarify why the "2 synapses or less" observation represents a significant difference between the hemilineage temporal cohorts compared to less related neurons.

See above comment for the changes.

B. The authors can choose to either:1. Perform:– Spatio-temporally controlled Notch loss and gain of function analyses on an example set of neurons (e.g. in one typical hemilineage);– Live imaging of how and when neurons invaded a target space and establish synapses

We performed Notch overexpression *spatially* restricted to single lineages (Figure 3); to also *temporally* restrict Notch e.g. to GMCs or mature neurons could be done by adding tsGal80 to the genotype, but would take months to build the genetics and do the experiment, or longer due to covid restrictions.

The second experiment, live imaging of neurite outgrowth and synapse formation, would be an incredibly difficult and require us to generate a genotype with UAS-myr:GFP+ lineage marker for membranes, UAS-RFP+ Brp puncta for presynapses, and UAS-Notch-intra for inducing ectopic dorsal projecting neurons; and even this genotype would not distinguish endogenous vs ectopic dorsal projections. Honestly, we don’t know of a genotype where we could selectively live image only the ectopic dorsal projecting neurons. And live imaging of the deep projections for several hours is probably at the very edge of feasibility, perhaps only possible with SCAPE microscopy (that we don’t possess).

Or:2. Tone down their conclusions about:– How Notch activity regulates targeting;– The extent to which hemilineage temporal cohorts correlate with developmental proximity.

Yes, we agree with your comment about Notch function. We add text in the discussion to indicate that we can’t distinguish the role of Notch overexpression in sibling cell fate from a role in post-mitotic neurons. Yet our findings are strongly supported by similar Notch overexpression experiments in larval neuroblasts which show that Notch misexpression transforms NotchOFF neurons to NotchON neurons in all tested lineages. Nevertheless, we are glad the reviewer raised the point, and we add a Discussion section addressing the unresolved issue of post-mitotic neuron Notch function: “Another point to consider is the potential role of Notch in post-mitotic neurons (Crowner et al., 2003), as our experiments generated Notch^intra^ misexpression in both new-born sibling neurons as well as mature post-mitotic neurons. Future work manipulating Notch levels specifically in mature post-mitotic neurons undergoing process outgrowth will be needed to answer this question.”

We are not sure what “developmental proximity” means – but if it refers to spatial proximity of hemilineage temporal cohort synapses, we agree. This is particularly true for postsynapse position (e.g. Figure 6, supplement 2). We note in the text “When we expand this analysis to all hemilineages, we found that (a) hemilineage-temporal cohorts had more similar synaptic positions than hemilineage alone, and (b) dorsal hemilineage-temporal cohorts preferentially clustered presynapses, whereas ventral hemilineage-temporal cohorts preferentially clustered postsynapses (Figure 6C-F).” We have also gone through the manuscript to ensure that we limit our conclusions on this point to the observed correlations. We leave to the discussion how the observed correlations lead to hypotheses about functional consequences, which need to be determined in a future study.